# Assessing performance and seasonal bias of pollen-based climate reconstructions in a perfect model world

Kira Rehfeld[1], Mathias Trachsel[2], Richard J. Telford[2,3], and Thomas Laepple[1]

[1]Alfred-Wegener-Institut Helmholtz-Zentrum für Polar- und Meeresforschung, 14473 Potsdam, Germany
[2]Department of Biology, University of Bergen, Postboks 7803, N-5020 Bergen
[3]Bjerknes Center for Climate Research, Allégaten 55, N-5007 Bergen, Norway

*Correspondence to:* K. Rehfeld[*] (kira.rehfeld@awi.de) and M. Trachsel[†] (mtrachs@umd.edu)

**Abstract.** Reconstructions of summer, winter or annual mean temperatures based on the species composition of bio-indicators such as pollen, foraminifera or chironomids are routinely used in climate model-proxy data comparison studies. Most reconstruction algorithms exploit the joint distribution of modern spatial climate and species distribution for the development of the reconstructions. They rely on the space-for-time substitution and the specific assumption that environmental variables other than those reconstructed are not be important, or that their relationship with the reconstructed variable(s) should be the same in the past as in the modern spatial calibration dataset.

Here we test the implications of this 'correlative uniformitarianism' assumption on climate reconstructions in an ideal model world, in which climate and vegetation are known at all times. The alternate reality is a climate simulation of the last 6000 years with dynamic vegetation. Transient changes of plant functional types are considered as surrogate pollen counts, and allow to establish, apply and evaluate transfer functions in the modeled world.

We find that in our model experiments the transfer function cross-validation $r^2$ is of limited use to identify reconstructible climate variables, as it only relies on the modern spatial climate/vegetation relationship. However, ordination approaches that assess the amount of fossil vegetation variance explained by the reconstructions are promising. We furthermore show that correlations between climate variables in the modern climate/vegetation relationship are systematically extended into the reconstructions. Summer temperatures, the most prominent driving variable for modeled vegetation change in the Northern Hemisphere, are accurately reconstructed. However, the amplitude of the model winter and mean annual temperature cooling between the mid-Holocene and present day is overestimated, and similar to the summer trend in magnitude.

This effect occurs because temporal changes of a dominant climate variable, such as summer temperatures in the model's Arctic, are imprinted on a less important variable, leading to reconstructions biased towards the dominant variable's trends. Our results, although based on a model vegetation that is inevitably simpler than reality, indicate that reconstructions of multiple climate variables based on modern spatial bio-indicator datasets should be treated with caution. Expert knowledge on the eco-physiological drivers of the proxies, and statistical methods that go beyond the cross-validation on modern calibration datasets are crucial to avoid misinterpretation.

[*]present address: British Antarctic Survey, Cambridge, UK
[†]present address: Department of Geology, University of Maryland, US

# 1  Introduction

Continental-scale climate reconstructions (Bartlein et al., 2011; Davis et al., 2003; Mauri et al., 2014) are frequently used as a paleo-data target to evaluate and benchmark climate models (e.g. Harrison et al., 2014; Fischer and Jungclaus, 2011). Currently, climate models and proxy data disagree on the annual mean temperature changes over the course of the Holocene (Liu et al., 2014; Marcott et al., 2013). It was argued that seasonal biases in proxy based climate reconstructions might be the root of the observed proxy-model divergence (Liu et al., 2014).

To arrive at quantitative assessments of past climate changes from pollen assemblages, transfer function algorithms are used to establish a link between modern climate and vegetation composition across space. The derived relationships are then applied to fossil pollen percentages, counted in sediment archives. A basic assumption underlying these transfer functions is methodological uniformitarianism (Scott, 1963; Gould, 1965), namely that modern spatial relationships between species, vegetation and environmental conditions can be applied to past conditions (e.g. Birks et al., 2010).

One specific requirement is that environmental variables other than those considered in the calibration are not important, or that their relationship with the reconstructed variable(s) was the same in the past as it is in the modern spatial calibration dataset (Birks and Seppä, 2005; Birks et al., 2010). Biological proxies generally respond to a multitude of environmental variables and thus the first part of the assumption is rarely met (Juggins, 2013). Therefore constancy and equivalence of the covariance of relevant parameters in space and time have to be assumed to allow the substitution of spatial gradients (in a modern calibration) for temporal changes (in the past) (Blois et al., 2013; Juggins, 2013). This assumption, which we name 'correlative uniformitarianism', is certainly violated in the real world. For example in the modern climate summer and winter temperatures are highly correlated across space. In contrast, the major driving forces behind the Holocene temperature evolution, local summer and winter insolation, have been anticorrelated over the past 10000 years due to precessional forcing (e.g. Laepple and Lohmann, 2009).

The validity of assuming correlative uniformitarianism, specifically the effect of confounding variables on reconstructions from bio-indicators, was investigated using simulated artificial data and it was shown that this can lead to misleading reconstructions and an underestimation of the prediction error (Juggins, 2013). However, without knowing the past climate evolution, it is difficult to estimate the potential implications for reconstructing the Holocene climate evolution.

Here, we use a Holocene climate model simulation with interactive vegetation as a testbed for pollen transfer function methods. In the model world, the modern spatial climate and its relationship to vegetation is known, along with the Holocene climate and vegetation evolution. Our general approach bears some similarities to previous 'pseudoproxy' experiments, where climate model simulations were used to test calibrations for temperature reconstructions of the last millennia (Mann and Rutherford, 2005; Küttel et al., 2007; von Storch et al., 2004). However, as these studies target proxy records for climate which are calibrated temporally against meteorological data (such as tree ring parameters), they largely focus on the effect of proxy noise on the reconstruction. We ignore these proxy imperfections and age uncertainty, and focus on the implications of correlative uniformitarianism, which is one operational assumption behind the use of spatial calibrations to reconstruct temporal changes. Key questions are: (i) To what extent does the correlative uniformitarianism, and aspects of the estimation processes, bias

reconstructions of the Holocene temperature evolution? (ii) Are there statistical indicators that can inform us about the actual reconstructability of climate variables?

To address these questions within the model world, we need to assume that model climate and vegetation changes are consistent with each other, and that modeled plant functional type (PFT) and land cover type changes (desert fraction) can be used as surrogates for pollen counts in sedimentary archives.

## 2 Methods

### 2.1 Climate model simulations

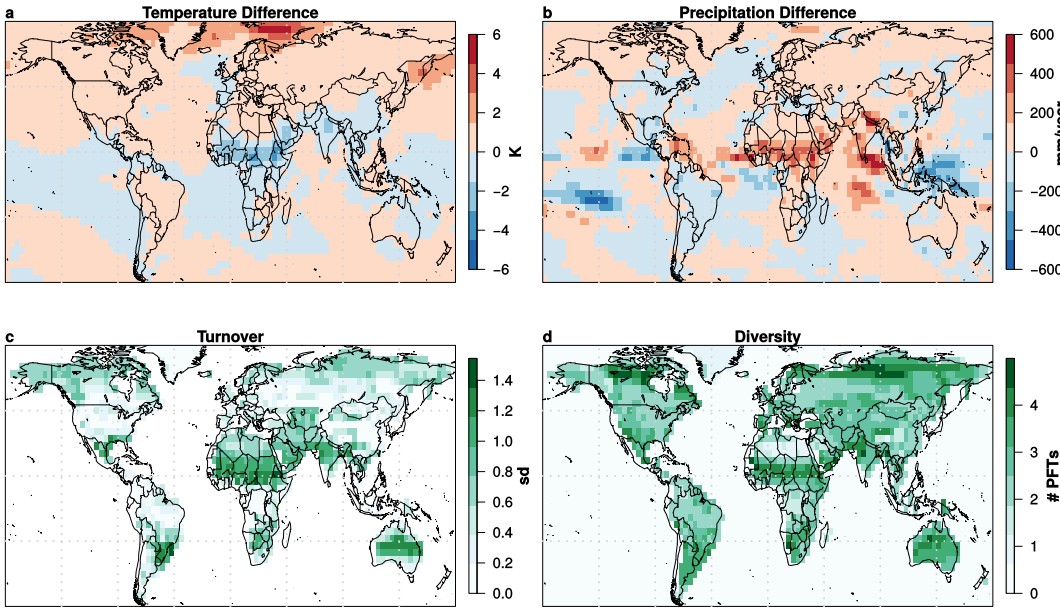

**Figure 1.** Temperature (a) and precipitation changes (b), vegetation turnover (c) and vegetation diversity as measured by the Hill's number $N_2$ of PFTs (d) between 6k and 0k BP in the ECHAM5/MPIOM model simulation (Fischer and Jungclaus, 2011).

We use a 6000-year-long transient simulation of the coupled atmosphere-ocean climate model ECHAM5/MPIOM (Jungclaus et al., 2006) with a dynamic land surface and vegetation scheme provided by the JSBACH module (Raddatz et al., 2007; Brovkin et al., 2009) to investigate pollen-based climate reconstruction techniques. This simulation is described in (Fischer and Jungclaus, 2011) (hereafter *6k-run*) and is only forced by orbital changes over the last 6000 years. Environmental and atmospheric variables are available on a regular $3.75° \times 3.75°$ latitude/longitude grid.

The vegetation module is described in Sitch et al. (2003) and Brovkin et al. (2009). The modeled climate-vegetation interaction through the growth, competition and mortality of the four tree, two shrub and two grass PFTs is nontrivial: Within each

grid cell, plants compete for fractional cover, given their own net primary productivity, natural mortality as well as disturbance-driven mortality in response to climate (fire, heat and cold extremes, growing season length). Given a latitude, soil texture, $CO_2$ concentration, temperature and precipitation, processes changing water balance, photosynthesis, leaf cover and respiration are simulated on a daily or monthly time step. The turnover of wood, leaves and roots, decomposition, mortality and establishment

is calculated annually, and the resulting vegetation cover is fed into the next year. Table 1 in the supplementary information lists the PFTs and their bioclimatic temperature limits.

The Holocene climate and vegetation evolution of this model simulation have been extensively used and characterized in paleoclimate model-data comparisons (Fischer and Jungclaus, 2011; Dallmeyer et al., 2011, 2013, 2015; Laepple and Huybers, 2014; Rehfeld and Laepple, 2016). While vegetation biases have been observed against present-day conditions in some areas

(Brovkin et al., 2009; Dallmeyer et al., 2011), the overall patterns are consistent (Brovkin et al., 2009). Climate and vegetation changes from mid-Holocene to present day are substantial (Fig.1) and differ between the seasons (Fig.3, top row). We note that, although the resolution of the climate model, and thus the model world calibration dataset is coarse, its spatial and seasonal range is comparable to that of real-world calibration datasets (SFig.1).

## 2.2 Reconstruction methods

Quantitative climate reconstruction (Juggins and Birks, 2012; Birks et al., 2010) based on a multivariate pollen count dataset requires algorithms that translate past vegetation changes into estimates of past climate changes. Most approaches use three datasets: A paired calibration set, and one downcore pollen record. The calibration set combines modern pollen and climate data from recent, or modern, conditions taken from surface samples across ecological and climatic gradients. An example from the real-world would be pollen counts from lake sediment surfaces across Europe, paired with data from meteorological

stations near these lakes. Several approaches for quantitative reconstructions based on ecological species counts have been established (see e.g. Birks et al., 2010, for a review). Here we focus on two popular techniques: Best Modern Analog methods (here: BMA, often also called Modern Analogue Technique), and the multivariate calibration method of Weighted Averaging (WA).

BMA methods directly match the species composition of fossil assemblages against the modern calibration set (Overpeck et al.,

1985). To obtain a reconstruction value for a fossil sample, $N$ analog modern samples with the lowest ecological distance (most commonly estimated using the Squared-Chord-Distance (Overpeck et al., 1985)) are selected. Their modern reference climate variables are averaged to obtain the past climate estimate. These approaches are expected to work well on samples with a low number of taxa. In this study we use BMA with N = 5 and the Squared-Chord distance.

Multivariate calibrations, on the other hand, are based on the regression of modern vegetation onto estimates of a climate vari-

able at many calibration sites, to establish one global parametric function between them. In WA calibration, climate optima for different taxa are derived by computing a weighted average of climate variable estimates at all sites at which a taxon is present. Weights are derived from the relative abundance of the taxon. The step from past vegetation composition to estimates of past climate then relies on a second weighting step, in which the climate optima of all taxa present in the fossil sample are averaged, again weighted by their relative abundance. We employ WA here to illustrate results that are common to reconstructions

based on BMA and WA-related methods, which may therefore depend on properties of the dataset, or the general approach of reconstructing climate based on modern spatial climate calibrations. In this study we use WA with square-root transformed scores and inverse deshrinking.

## 2.3  Estimates of reconstruction uncertainty

In a real-world situation, the true climate evolution is unknown and a root mean square error of prediction (RMSEP) is estimated in the modern calibration set. In the following we use k-fold cross-validation with k=10 (1/k-th of the samples are used for verification) but note that even using leave-group-out-cross-validation, the RMSEP may be biased low due to autocorrelation in the modern data (Telford and Birks, 2005, 2009). As we know the true climate in the model world, we can additionally obtain the root mean square error of the reconstruction (RMSE) by comparing the reconstructed climate variable to its simulated counterpart.

We employ multivariate constrained ordination methods to test which climate variables explain vegetation variance. While Redundancy Analysis (RDA) extends principal component analysis, Canonical Correspondence Analysis (CCA) is the equivalent method for frequency data and allows a unimodal relationship between the species and the environment (Borcard et al., 2011). We evaluate the similarity between trend and correlation fields using a sign-test, similar to Kendall's rank correlation, defined as a fraction $\nu(X,Y) = \frac{S(X,Y)}{\text{\# reconstr. grid cells}}$ varying between -1 and +1. A grid cell counts into the sign sum $S(X,Y)$ as +1 if the signs in field X and field Y are the same, and as -1 if they are opposite. Summation goes over all grid cells where a reconstruction was performed. This sign test yields $\nu = 1$ if and only if all grid cells in field X and Y have the same sign, and $\nu = -1$ if all signs are opposing. $\nu = 0$ suggests that there are as many grid cells with opposing signs as there are with the same signs, indicating that there is no underlying similarity between the fields.

## 2.4  Calibration and reconstruction workflow

We perform PFT-based calibrations and climate reconstructions at each grid point on land which displays enough diversity and temporal variations in the simulated vegetation. Therefore, we select all points for the reconstruction tests with an effective number of taxa $N_2$ larger than 2 (Hill, 1973)[3], and vegetation turnover larger than $0.5$. Turnover is estimated from the length of the first detrended correspondence analysis axis in standard deviation units (Hill and Gauch, 1980).

The simulated vegetation history through time at a grid point forms the fossil vegetation dataset. The simulated modern surrounding vegetation and climate fields, averaged over the last 30 years, yield the matrices containing modern pollen and climate information for the modern training set. We select all surrounding land-points in a radius of 2500km and subsample them such, that the calibration set size is roughly equal for all sites and not latitude-dependent.

Pollen matrix columns contain the percentages of the nine PFTs (acronyms in Appendix A, details in Suppl. Table 1), including the desert fraction as a virtual PFT. Each column in the modern climate matrix corresponds to a climate variable and we choose

---

[3]The Hill's number $N_2$ is defined as $N_2 = \left( \sum_{i=1}^{N} p_i^2 \right)^{-1}$, as the reciprocal of the weighted mean of the abundances $p$. If all taxa are equally abundant and $p_i = 1/N$, $N_2$ is equal to N. If only one taxon is present, and all others are zero, $N_2 = 1$.

the warmest month, coldest month and annual mean temperatures (MTWA, MTCO, MAT) and precipitation (MPWA, MPCO, MAP) variables.

We note that large-scale PFT-based pollen reconstructions use roughly 2-3 times the number of PFTs (as e.g. in Davis et al., 2003; Mauri et al., 2014), and raw pollen spectra contain often more than 10 times the number of taxa. However, the effective
number of PFTs in the fossil record, as estimated by Hill's $N_2$, is much lower than the number of taxa itself, and rare taxa do not have a large influence on reconstructions using BMA or WA. Our cutoff at $N_2 = 2$ is well within the range of $N_2$ for modern pollen spectra (SFig. 2), although the $N_2$ is lower for PFTs than for taxa by construction. In general, a low number of PFTs or taxa may lead to a problem of multiple analogs, where a pollen assemblage is similar to several modern assemblages that are very different in their climatic setting (ter Braak et al., 1996).
However, supporting our cutoff choice at $N_2 = 2$, we do not find indications that this is a problem here. The overall high transfer function $r^2$ (Fig. 7) shows that analogs are not picked at random from the training set. To pinpoint this further, we calculate the ratio of the standard deviations of the temperatures at the analog sites, and the standard deviation of the temperatures across the whole training set (SFig. 3). The ratios are generally smaller than 0.5, thus illustrating that the analog sites are not randomly drawn from the training set.
In many conventional paleoecological studies, one or two climate variables would be selected for reconstruction, which are expected to have influenced vegetation development significantly, and independently (Juggins, 2013; Telford and Birks, 2011). As we want to investigate, which variables can be skillfully reconstructed, we perform joint reconstructions of all six climate variables, both via BMA and WA. We note that jointly reconstructing several climate variables is done in several large-scale regional reconstructions (e.g. in Mauri et al., 2014; Bartlein et al., 2011; Davis et al., 2003) and come back to this later in the
discussion.

Fig. 2 illustrates the whole calibration and reconstruction workflow for a BMA reconstruction at an example grid point selected from the Arctic (120°E,72°N). CCA analyses (Fig. 2d) suggest, that summer temperature is the main climate variable driving modern vegetation around the site, whereas winter temperatures have little to no impact on the vegetation changes in the model. A summer temperature calibration based on BMA can explain considerable amounts of variance in the modern
vegetation-climate relationship, it also shows a low RMSEP of $\sim 1.15°$ C. In the model world, we can compare reconstructed and the simulated true past model climate evolution (Fig. 2f) and find that summer temperatures (MTWA) are faithfully reconstructed, whereas the reconstructions of annual mean (MAT) and winter temperatures (MTCO) largely fail.

## 3   Results

### 3.1   Simulated and reconstructed Holocene temperature trends

The simulated mid-late Holocene temperature evolution shows a zonal structure characterized by warming trends around the Equator and across Asia and cooling trends in the mid-to-high latitudes (Fig. 3 top row). The seasonal insolation forcing caused by changes of the orbital configuration results in distinct temporal trends for summer and winter temperature, which differ in their strength and in some regions also in their signs. In the Arctic regions, the trends in the model simulation are strong ($\sim$-

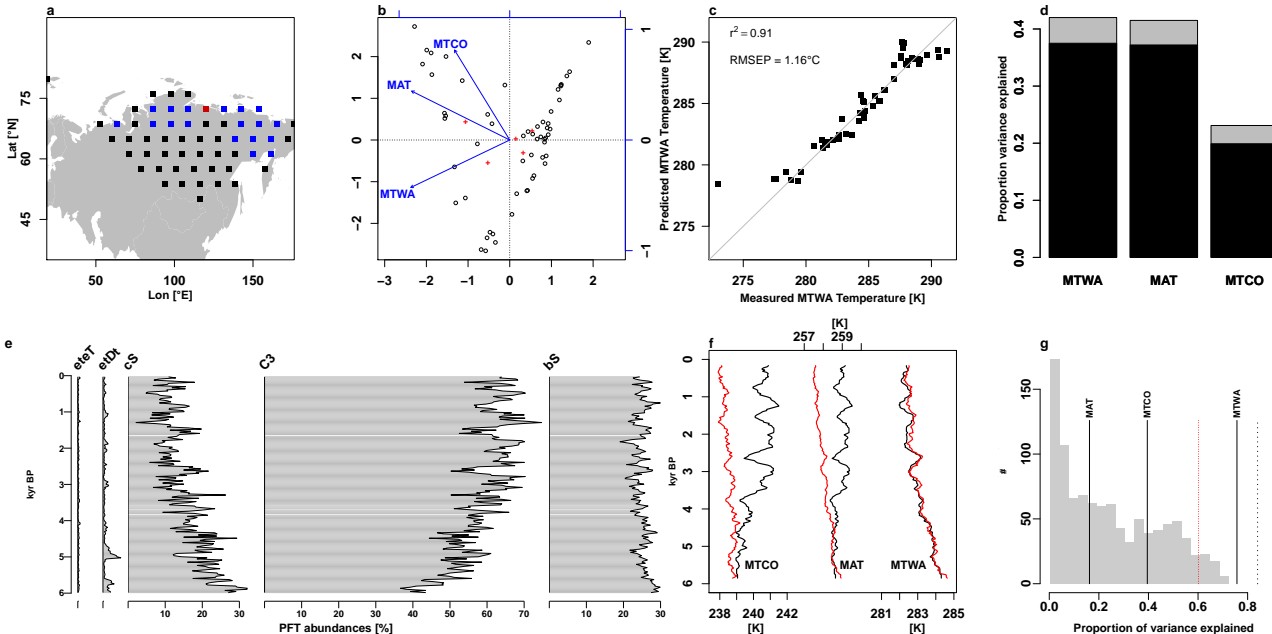

**Figure 2.** Exemplary calibration, BMA reconstruction and verification workflow for the grid point site in Siberia (120°E,72°N) highlighted as a red square in (a). Surrounding grid points from which the modern analogs are drawn are shown as black dots, chosen analogs in blue. CCA analyses show that MTWA explains most variance in modern vegetation (b&d) and performs sufficiently well in leave-one-out cross validation (c). The jointly reconstructed climate variables show considerable shared (black), and rather little independent variance (grey) in the modern calibration (d). Past vegetation changes, as shown in the percentage PFT diagram (e), appear to be correlated with (f) simulated and reconstructed climate. PFT acronyms are listed in the appendix. Red lines show the simulated 'true' past temperatures, black lines the reconstructions. (g) The MTWA reconstruction explains most fossil vegetation variance in the `randomTF` significance test, compared to the other temperature variables, and falls outside the confidence interval of the test (red line). The dashed line corresponds to the maximum amount of variance a single variable can explain (Telford and Birks, 2011).

0.5K/kyr) for summer, and weaker ($\sim -0.1$K/kyr) for winter and the annual mean. The warming trends around the Equator appear strongest in the coldest month. Similar patterns occur in the mean annual precipitation, with drying in the Northern and wetting in the Southern Hemisphere. We focus here on temperature and refer the reader with interest in the precipitation changes to SFig. 4.

5   We now analyze the winter (MTCO), summer (MTWA) and annual mean (MAT) temperature patterns reconstructed using BMA and WA (Fig. 3 middle & bottom rows). Reconstructed winter trend patterns diverge from the simulated trends. In many regions the reconstructed trends are higher than $\pm 1$K/kyr in magnitude, and thus stronger than anywhere in the simulated model climate. Negative temperature trends in polar regions are not consistently captured, and an east-to-west warm-to-cold gradient appears for both reconstruction techniques WA and BMA.

10  In contrast, the reconstructed summer trends show broad similarities to the simulated temperature changes. Equatorial warming

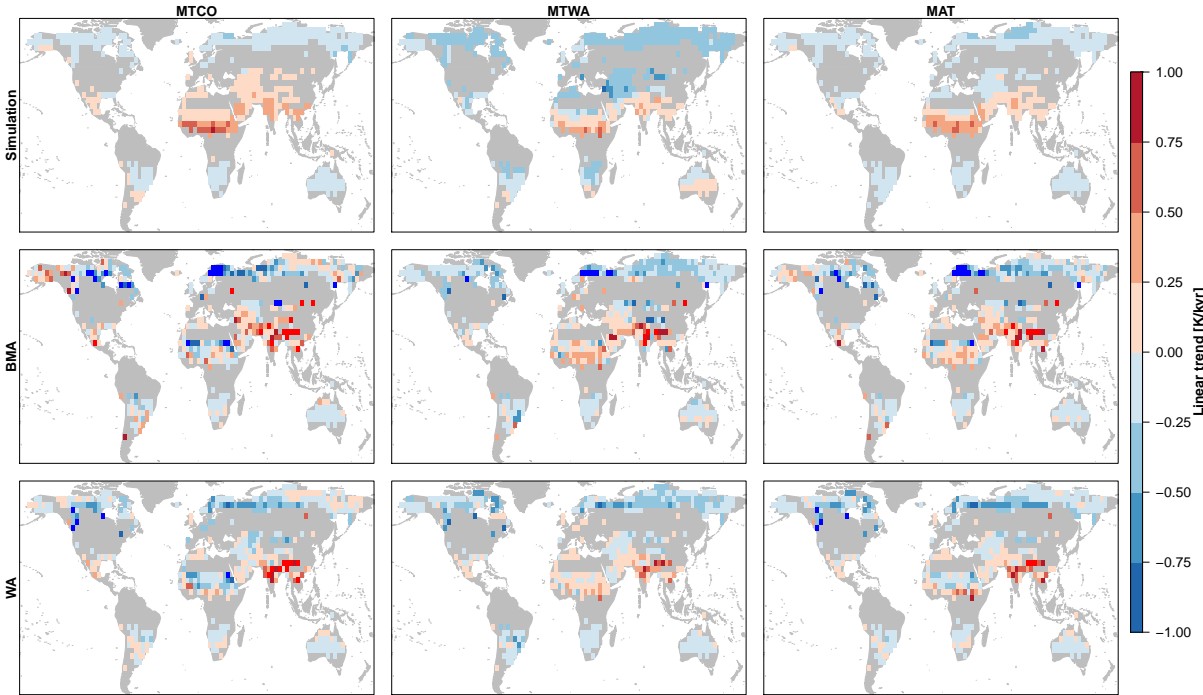

**Figure 3.** Linear trend in the simulated (top row) vs. the reconstructed temperature evolution between 6k and present day based on BMA (middle row) and WA (bottom row). Saturated red/blue colors indicate that the grid point's trends are stronger than 1K/kyr.

and polar cooling are captured by both WA and BMA. Differences exist in the magnitude of the changes, rather than the sign, except for in the Middle East, where warming is suggested by BMA and WA, and the true simulation trends showed a cooling, in particular around present-day Turkey.

Amongst the climate variables, MTWA appears to be most consistent between simulations and reconstructions. This is also

5  supported by the results of the sign test (described in Sec. 2.3), which yields $\nu \approx 0.5$ for WA and BMA. MTCO is least consistent ($\nu \approx 0.3$). Between WA and BMA, results appear more patchy for BMA than for WA (i.e. sign or magnitude vary less gradually across space), but this does not imply that either method captures correct degrees of change. This is further underlined by the temperature standard deviations taken across the trend fields, which are much larger for WA ($\overline{sd} = 1.8$K, bottom row in Fig. 3) and BMA ($\overline{sd} = 2.9$K, middle row) than for the simulation ($\overline{sd} = 1.2$K, top row). Thus, for both reconstruction methods

10  reconstructed trends are spatially more heterogeneous than the simulated trends.

The spatial patterns and magnitudes of the reconstructed trends are very similar across all three seasons (compare panels across rows in Fig.3). Visually, they show a stronger similarity than the spatial patterns of the simulated seasonal trends (compare panels of the top row). This is due to the fact that grid cells with large positive or negative trends appear in the same positions across the seasons (i.e., row-wise), but not necessarily across methods (i.e., column-wise). The sign test shows slightly larger

15  correspondences within each row/across seasons for the same method ($\overline{\nu} = 0.59$) than for the columns/same season across

methods ($\overline{\nu} = 0.47$). Due to the influence of the strong trends in the same places, this discrepancy is stronger for Pearson correlations across the fields of Fig. 3 (by method $\overline{\rho} = 0.79$, by season ($\overline{\rho} = 0.46$). One explanation for this observation could be that all seasonal reconstructions are biased towards a single specific season.

## 3.2 Seasonal bias of temperature reconstructions

To further investigate this finding, we analyze the correlation between the different seasons in the simulations across modern space and across time and contrast them with the correlation through time between the reconstructed seasonal time series (Fig. 4). Ideally, the temporal correlation of the reconstructions should equal the temporal correlation of our 'true' (model simulated) climate evolution. Correlations across modern space are calculated over all the grid points relevant in the calibration and reconstruction process, thus for WA these are all grid boxes in a radius of 2500km whereas for BMA, only the sites picked as modern analog in the reconstruction are used (see Fig. 2a for an example). For simplicity, we perform the analysis for winter (MTCO) against summer (MTWA) temperature, but other variable combinations (e.g. temperature against precipitation) would lead to similar results.

Across modern space MTCO and MTWA are mostly positively correlated (Fig. 4a), as towards the poles temperatures get colder in summers as well as in winter. Exceptions are found around Eastern Russia and equatorial regions in Africa, where summer and winter temperatures are anti-correlated across space.

The temporal correlations of the WA-reconstructed MTCO and MTWA (Fig. 4b) show a very similar pattern of the correlation sign, although with stronger amplitudes of the correlation values. Indeed, the sign test yields $\nu = 0.76$, indicating that the large majority of the grid cells in Fig. 4a and Fig. 4b share the same sign. In contrast, the 'true' temporal MTCO/MTWA correlation over the late Holocene (Fig. 4e), which should ideally be similar to the reconstructed temporal correlation (Fig. 4b), shows a different picture ($\nu = 0.26$). This suggests that the modern spatial covariance has been directly propagated to the temporal covariance of the reconstructions. Here, and in Fig. 5, we mask grid points for fossil reconstructions with low transfer function performance as measured by the cross-validation r$^2$, as we expect them to return less reliable results.

The same observation holds for the BMA-based results (Fig. 4d). The modern spatial MTCO/MTWA covariances at the sites picked as modern analogs, shown in Fig. 4c, are noisier than the covariances calculated over all grid boxes, but show a similar pattern. The seasonal correlation in the BMA-reconstructions again directly follows the modern spatial MTCO/MTWA correlation ($\nu = 0.68$). In contrast, the similarity to the actual temporal covariance (Fig. 4e) is low, as the sign test underlines ($\nu = 0.03$).

## 3.3 Reconstruction skill

We showed that the ability to reconstruct Holocene temperature trends in our model world strongly depends on the analyzed season and region (Fig. 3). It is also important to quantify the reconstruction skill for the full Holocene evolution, including millennial variability and absolute temperature estimates. We analyze two metrics, (i) the temporal Pearson correlation between the 'true' past changes and the climate variable reconstructions ("correlation skill", Fig. 5), and (ii) the RMSE deviation of the

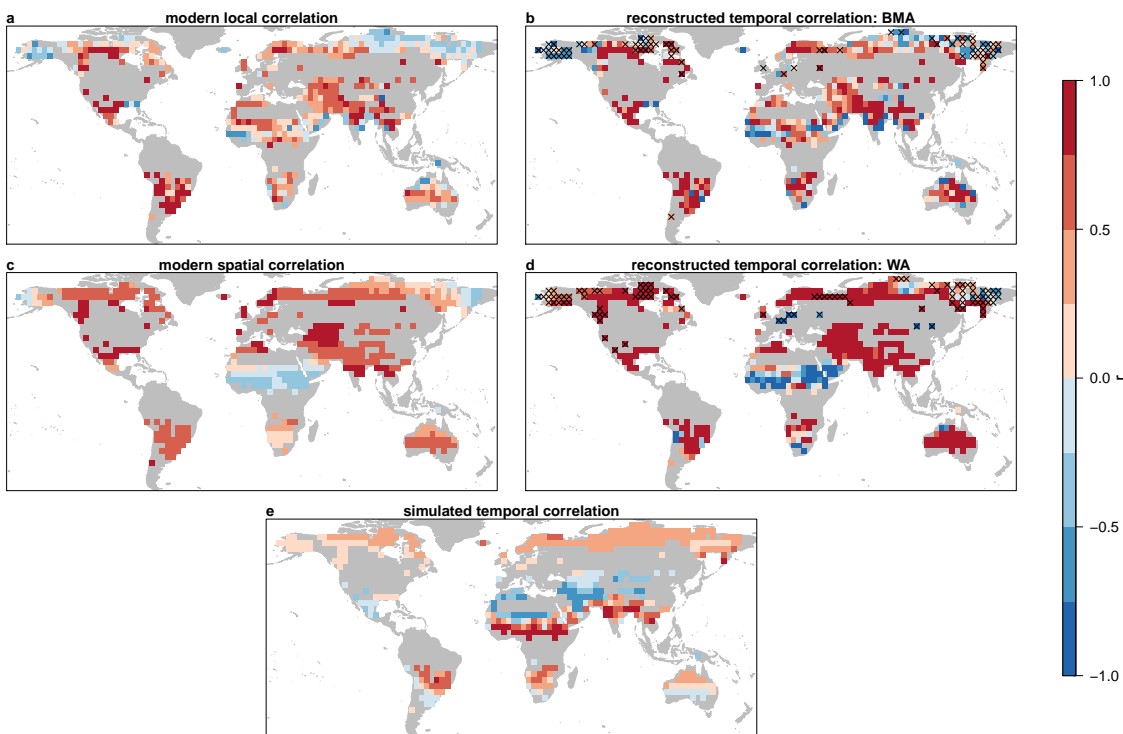

**Figure 4.** Correlation of coldest and warmest month temperatures. The correlation patterns across modern calibration space (a) are similar to the temporal correlation pattern estimated from WA reconstructions (b). The correlations at the sites picked as modern analogs (c) are similar to those obtained in the final BMA reconstructions (d). In contrast, the 'true' temporal correlation pattern from the model temperatures differs considerably from the reconstructed temporal correlation fields. This demonstrates that the correlation in the reconstructions mainly depends on the modern calibration and not, as one would hope for, from the correlation of the Holocene temperature evolution. Crosses in (b) and (d) indicate gridboxes with a $r^2 < 0.5$ in cross-validation.

reconstructed from the 'true' climate.

Consistently high correlation skill values for the BMA reconstruction can be found across the Arctic for MTWA, and in the Sahel for MAP. Simulated MAT changes are correlated with MTWA changes in the high latitudes, which explains the relatively weaker but positive correlation there. Winter precipitation reconstructions do not show good skill anywhere.

5    Most regions with high positive correlation skill show comparably low temporal RMSE (SFig. 5), whereas many regions with low RMSE do not show high correlation skill. In a real-world situation, the true past climate evolution is unknown and a root mean square error of prediction (RMSEP) is estimated from the modern calibration set (cf. Sec. 2.3). In our model world, the RMSEP is below 3°C for MTWA and MAT, whereas it is generally higher for winter temperature, in particular for North America. The low correlation skill for winter temperatures in the Arctic is also reflected by the temporal RMSE and the modern

10    RMSEP (SFig. 5 & 6). A comparison of summer temperature downcore RMSE and modern spatial RMSEP, given in SFig. 7, shows that modern RMSEP is higher than the actual reconstruction error in many places, but there is little resemblance to the

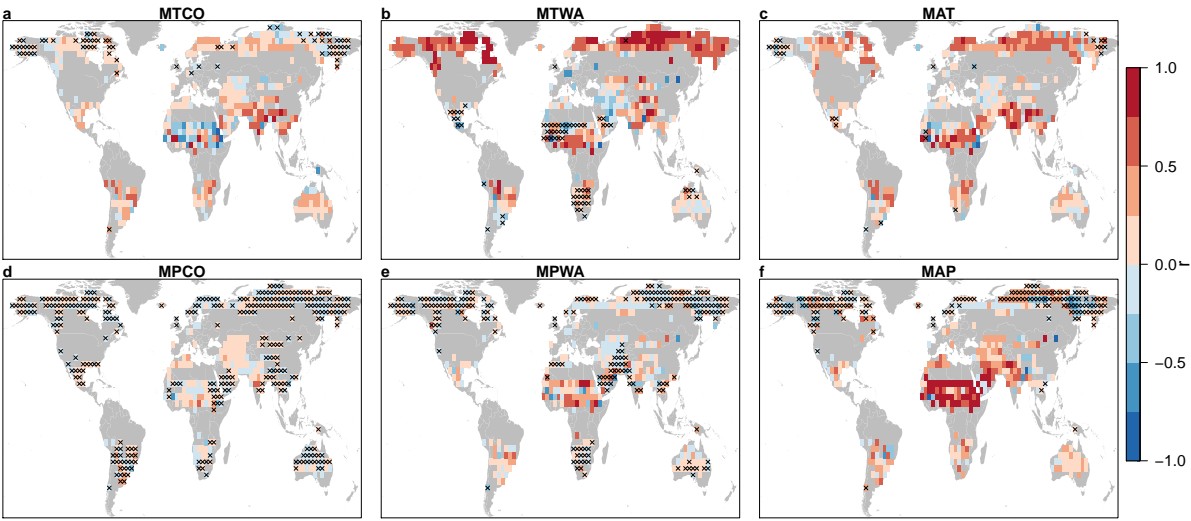

**Figure 5.** Performance of the BMA calibration models as evaluated by the correlation between the reconstructed and simulated climate variables (a-f) at each grid point. Crosses mask grid boxes with cross-validation $r^2 < 0.5$.

patterns of the estimated downcore RMSEP. If the calibration radius is reduced, the modern calibration error decreases (results not shown).

## 3.4 Testing for the predictability of reconstruction skill

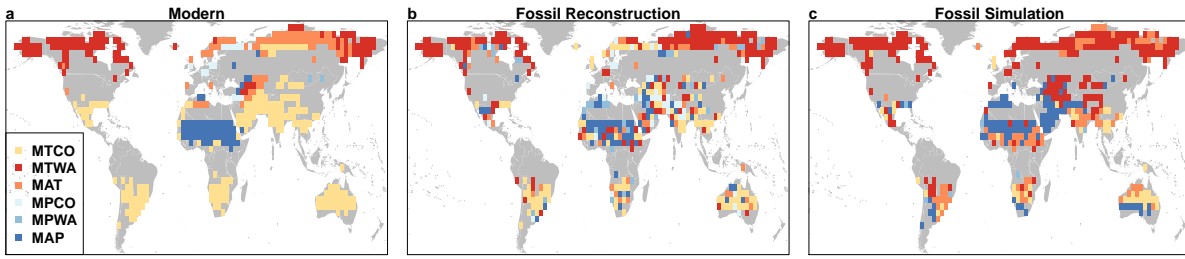

**Figure 6.** Climate variables explaining most variance in modern vegetation (a), between reconstructed climate and fossil vegetation (b) and simulated climate and fossil vegetation (c). Variables explaining most variance in the modern world (a) are not necessarily those explaining vegetation changes in the 'true' model past (c).

The inaccuracy of the covariance estimates (Fig. 4b), and the dependency of the reconstruction skill on the analyzed climate variable (Fig. 5) highlights, that it is important to determine which climate variables can be reconstructed in a given setting - and what other variables they are colinear with in the modern training set. We can discern two statistical approaches to identify the driving variable for climate-related vegetation changes: Those relying on the modern calibration set, and those which

involve the fossil downcore record. In both, higher variance explained should be reflecting a higher environmental relevance (Juggins and Birks, 2012).

In the following, we compare the results of estimating the driving climate variable with both approaches (Fig. 6a,b), with the pattern of the 'true' climate variable explaining most simulated fossil vegetation change in our model simulation (Fig. 6c). The

ordination fields underlying this summary figure are given in the SFigs. 8 to 10. For the modern spatial approach, we use CCA ordination of modern PFTs and climate to determine the climate variable which explains most vegetation variance across the modern calibration space (Fig. 6a). Temperature variables dominate the ordination results globally, except for the Sahel zone, which is dominated by precipitation-changes. MTWA explains most variance in arctic Canada and eastern Siberia, whereas MAT appears to dominate in Siberia and Northern Europe.

For the fossil downcore record approach, we identify which BMA-reconstructed climate variable explains most variance in the fossil vegetation set using constrained ordination (RDA). The results, as can be seen in Fig. 6b, are different and less smooth than those obtained for the modern spatial vegetation changes. Note that the patterns we observe here are highly similar to those identified from the ratio of the first two axes of the ordination (Juggins, 2013) (SFig. 11).

Finally, as we have access to the 'true' past vegetation and climate changes in the model world, we can assess, which climate

variable explains most simulated fossil vegetation change. The RDA results, shown in Fig. 6c, confirm a strong summer temperature signal above the Arctic circle, and the potential existence of a precipitation signal in the Middle East and the Sahel zone.

Contemplating Fig.6a, b, and c we observe that the driving variables, identified by the fossil downcore approach (Fig. 6b) are closer to the true (Fig. 6c) driving variables than the driving variables estimated from the modern calibration dataset (Fig.6a).

This suggests that looking at the variance explained by downcore reconstructions may tell us more about what actually drove vegetation changes, than looking at the variance explained in modern vegetation.

Furthermore, analyzing the variance explained in the modern calibration dataset can suggest a high importance (by a high explained variance) for variables that are not necessarily relevant to vegetation development. This is due to the colinearity of the climate variables (c.f. Fig. 2b). This is demonstrated in Fig. 7, which shows the transfer function $r^2$ for all climate variables.

In large parts of Siberia, MAT explained most variance (Fig. 6a). However, MTWA transfer function $r^2$ (Fig. 7b) is about as high as that of MAT (Fig. 7c) there, and dominates the rest of the Arctic. MAP appears well reconstructible in the Southern Hemisphere, in regions where MTCO also has a high transfer function $r^2$. Seasonal precipitation transfer functions do not perform well on inter-regional scales outside Africa. There, they appear to perform better, which is likely due to their colinearity with MAP (c.f. Fig. 6).

For the potentially more skillful approach of using the downcore reconstruction to test for reconstruction skill, a formalized test (`randomTF`) has been proposed in Telford and Birks (2011). It relies on the comparison between the fossil variance explained by the actual reconstruction, and the variance explained by reconstructions based on surrogate modern climate (but using the same modern and fossil pollen assemblages). Above 50°N, where temperature changes occur over the course of the 6k-run, 84.7% of the grid cell vegetation changes are identified as most strongly related to MTWA (Table 1). If the `randomTF`-test has

power, it should indicate a lower p-value for reconstructions of climate variables that were related to vegetation changes. Table

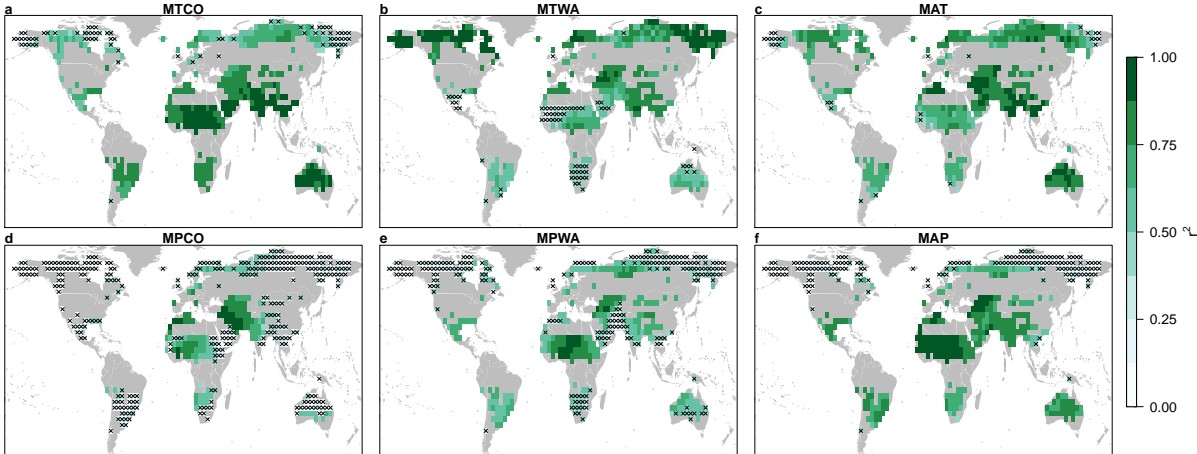

**Figure 7.** Spatial patterns of BMA transfer function $r^2$ in the modern calibration set (grid points with a distance of less than 2500km from the reconstruction site) of the six jointly reconstructed climate variables MTCO (a), MTWA (b), MAT (c), MPCO (d), MPWA (e), MAP (f). Points with a $r^2 < 0.5$ are crossed out. Transfer function performance appears good, although some variables had little impact on vegetation changes in the past.

indicates a significant p-value ($\leq 0.1$) for MTWA in 68.9% of grid cells. MAT, picked as most relevant in 14% of the grid cells, appears reconstructible in 23% of the grid cells. MTCO, MAP, MPCO and MPWA – which have no or little relevance for vegetation development in the region – show up as significant in only 14-16% of the grid cells. Although our test approach does not meet the criteria of a formal statistical power assessment, these results suggest, that `randomTF` may have indicative power.

**Table 1.** Outcome of the significance test using `randomTF`. All 196 grid points above 50°N are considered, and p-values are estimated for all climate variables. Actual relevance is obtained by counting the number of times the variable is picked as the most relevant variable in the RDA of simulated climate and vegetation (Fig. 6) and dividing by the number of grid cells.

| | Relevance [%] | `randomTF`: significant (p< 0.1) | | | `randomTF`: not significant (p> 0.1) | | |
|---|---|---|---|---|---|---|---|
| | | RMSEP | r(rec,sim) | No. cells [%] | RMSEP | r(rec,sim) | No. cells [%] |
| **MTCO [°C]** | 1.5 | 4.16 | 0.17 | 13.8 | 3.31 | 0.08 | 86.2 |
| **MTWA [°C]** | 84.7 | 0.92 | 0.71 | 68.9 | 2.00 | 0.37 | 31.1 |
| **MAT [°C]** | 13.8 | 2.43 | 0.56 | 23.5 | 2.13 | 0.26 | 76.5 |
| **MPCO [mm yr$^{-1}$]** | 0.0 | 180.80 | -0.03 | 9.2 | 113.63 | 0.00 | 90.8 |
| **MPWA [mm yr$^{-1}$]** | 0.0 | 237.44 | 0.06 | 16.3 | 184.9 | 0.00 | 83.7 |
| **MAP [mm yr$^{-1}$]** | 0.0 | 150.52 | 0.21 | 15.8 | 123.76 | 0.04 | 84.2 |

### 3.5 Influence of the modern climate background on the reconstructed climate

Ideally, a climate reconstruction should not depend on the climate state in which the calibration set was taken. We test this in a case study, by comparing the calibration to the most recent time period (the last 30 years of the model run, equivalent to 0-30yrs BP) which we use throughout the manuscript, to one for the first period (5970-6000 yrs BP) in the simulation. We subsequently perform reconstructions for both calibration periods. Fig. 8 shows exemplary BMA results for a Siberian site.

Averaged across all reconstruction sites, MTWA reconstructions calibrated at 6k are .75K (-3.6,1.7K, 90% confidence inter-

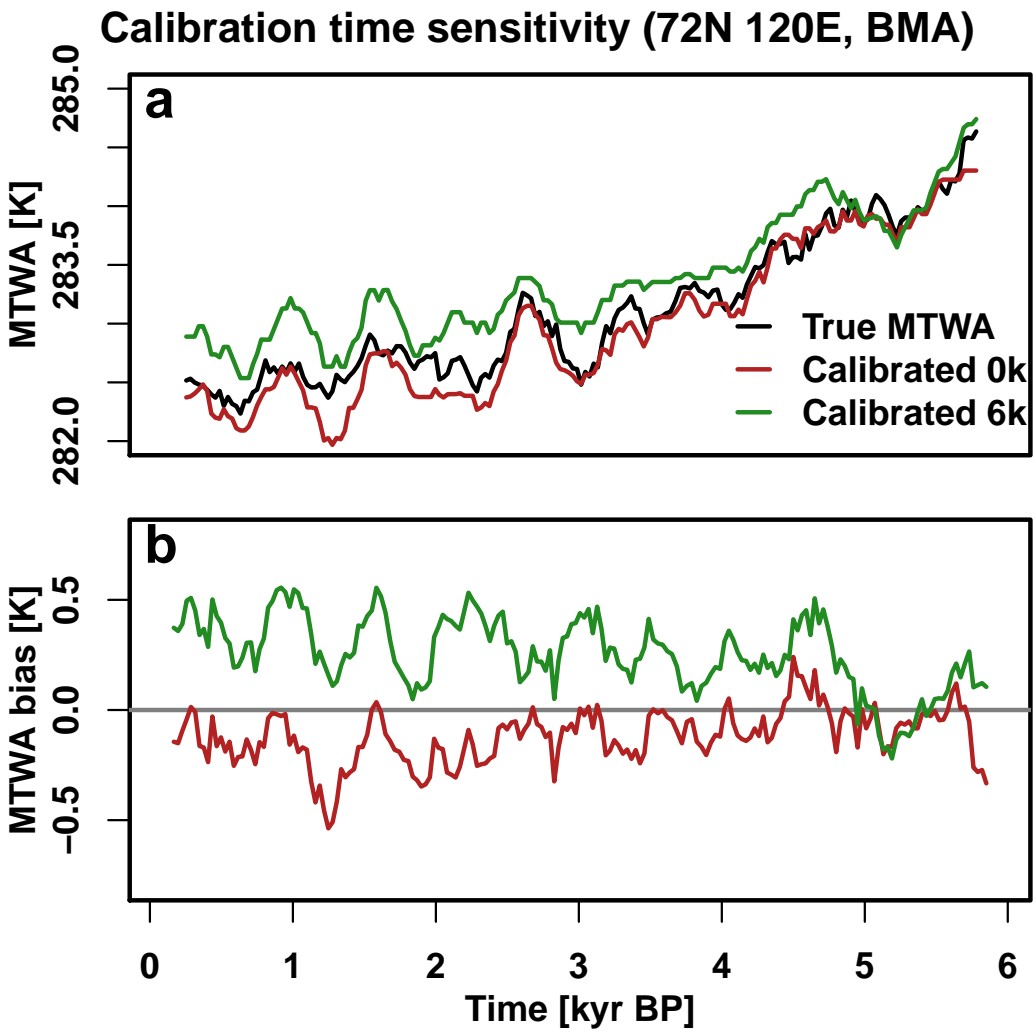

**Figure 8.** Reconstructions are sensitive to the calibration time period. Warmest month temperature trends for reconstructions based on a calibration for the last 30 years (0k) and first 30 years (6k) of the model run (A). 6k results are mostly warmer (B). All time series are based on 300-year running means.

val) warmer than those based on calibrations at 0k. In particular, sites across the Northern Hemisphere are reconstructed with warmer temperatures. Relative temperature variations largely match between the reconstructions. Inspection of the locations and temperatures around the analog sites chosen for the 0k and 6k calibrations suggests that the warm bias may be caused by spatial autocorrelation in the vegetation, rather than climate, in addition to other local confounding factors. The 6k analog sites tend to lie further northward (in the Northern Hemisphere) than those for the 0k calibration. However, the 6k analog sites do not systematically cluster northward. Therefore, the northward migration of the analog sites does not compensate fully for the warmer background climate state, so that the overall reconstructed temperatures are warmer. This demonstrates that, at least in our experiment, the climatological and ecological similarity of the calibration period to the period for reconstruction influences the reconstruction outcome.

The question whether the detected differences in Fig. 8 are significant or not using the calibration RMSEP is not straightforward. A standard assumption in paleoclimate reconstructions is that errors in time and space are independent (as assumed e.g. in Marcott et al., 2013). This assumption would result in a standard error of 0.13°C, thus considerably smaller than the differences we found. In the (unrealistic) extreme case of a complete dependency of errors, the differences would be not significant (standard error 3.5°C). In reality the true uncertainty likely lies between the two extremes assumed here, but a more detailed analysis of the spatial and temporal covariance structure of the proxy uncertainty is required to provide better error estimates.

## 4 Discussion

Using a Holocene climate model simulation as a testbed for pollen based climate reconstructions allowed us to analyze the reconstruction skill and to understand potential seasonal biases of pollen based climate reconstruction methods.

### 4.1 Correlative uniformitarianism

Transfer function reconstructions rely on the exchangeability of spatial and temporal relationships between climatic, environmental and ecological variables; on the the uniformity of correlations across space and time. We have demonstrated that spatial and temporal correlations are not equivalent on orbital timescales in our model world Holocene, which has implications for seasonal temperature reconstructions. The space-for-time substitution in transfer functions hence leads to seasonal biases in the reconstructions, as the assumption of correlative uniformitarianism is violated.

This is consistent with findings of Blois et al. (2013), who tested the space-for-time substitution for the prediction of biodiversity changes. They observed that while generalized dissimilarity models fitted across space could predict large-scale patterns of diversity across time through the late Quaternary, the relationship between turnover and environmental variables was different through space and through time. Furthermore, space-for-time substitution was less successful for the Holocene, which is likely due to the relatively smaller temporal climate variations compared to the spatial variations.

Salonen et al. (2013) showed that reconstructions using different modern calibration datasets differed in their means, and variations around this mean. The calibration datasets had different temperature distributions. This could be a consequence of a violation of correlative uniformitarianism: the relationships between climate variables and ecological changes, which are transferred to the final reconstruction, are likely different for calibrations extending to different locations (as in Salonen et al. (2013)), or for different time periods (as in Section 3.5).

## 4.2 Limitations

The complexity of the vegetation representation in the model, as well as the simulated climate evolution, are a strong simplification of reality. Therefore, results on the Holocene evolution of specific PFTs, the actual spatial pattern of PFTs, or the reconstructability of a certain climate variable in a certain region should not be directly translated to actual pollen-based climate reconstructions. On the other hand, conclusions on reconstruction methods and the relation of spatial calibration and downcore reconstruction only require a consistent dataset of climate and vegetation parameters in space and time and do not depend on details of the climate evolution or vegetation response, as long as the dataset is realistic enough that we can apply the PFT-based reconstruction workflow. The major factor shaping our results is that the modern spatial relationships between climate variables is different from the changes in the relationships over time, which is a robust feature related to the transient insolation forcing (Laepple and Lohmann, 2009).

One might be concerned that the low number of simulated plant functional types, or the low spatial resolution of the model, might bias our reconstruction efforts. However, we showed that the actual information contained in the plant functional types and the spatial climate field is not fundamentally different from that in the PFTs (or taxa) and the climate calibration datasets used in real world reconstructions (SFigs. 1 & 2). Note that it is likely, but not proven, that a larger Hill's $N_2$ ensures more meaningful reconstructions. What constitutes a too low number, or a significant difference in $N_2$, is as yet unknown.

Given the design of our study, we have limited our analyses to identifying general features of the calibration vs. reconstruction relationship, rather than interpreting the actual numbers of temperature changes or reconstruction biases. Furthermore, we assumed perfect proxy recording and did not add any non-climatic noise. If these were added, tests which rely on the downcore record, such as `randomTF`, may become less powerful, and downcore RMSE could become higher.

Our main study region - the Northern Hemisphere, and Arctic Russia in particular, is characterized by cold temperatures and is particularly sensitive to the orbital changes in the model simulation. Hence MTWA is the predominant driving variable. Multivariate analyses suggest that this is not the case everywhere (c.f. Fig. 6). Given the conceptual nature of our study - and the simplicity of the vegetation model - we have limited our discussion to the identifiability of a single driving variable. This does not exclude that in other regions multiple climatic controls on vegetation may be more important. Thus, other regions may be better suited to test the ability of transfer functions to disentangle changes in multiple climate variables.

## 4.3 Identification of climate variables driving vegetation evolution through time

Our study shows that in our model world, regardless of the reconstruction technique, the reconstructed climate evolution is very similar between the variables (Fig. 3). This strong covariance between the variables is determined by the modern spatial

covariance and not, as one would hope, the temporal covariance of local climate (Fig. 4). This finding can be understood in a simple thought experiment. Let us assume that the vegetation evolution at every grid point would be driven by one single variable. This single variable could be one of the analyzed variables (e.g. summer temperature) or any other variable, such as the length of the growing season, cloudiness or soil moisture. All other variables have no direct influence on the vegetation,

themselves, and are merely covarying with the driving variable. In this case, the reconstructed covariability is implicit in the transfer function and fully determined from the modern spatial relationship, regardless of the true past relationship between the variables, and this is similar to what we found (Figs. 3 and 4).

Reconstruction skill will consequently depend on whether we reconstruct the driving variable, or, in the case that we reconstruct a secondary variable, on whether the the relationship with the driving variable is the same across space and in time. The example

of our model-world Arctic shows that the latter is not always the case. Past vegetation changes there, as Fig. 6 shows, were predominantly driven by summer temperature and mean annual temperature change, yet the modern transfer function $r^2$ for MTCO is acceptable in most grid boxes (Fig. 7). Skill for winter temperature reconstructions are, however, low (Fig. 5a), particularly in regions where the modern spatial covariance between summer and winter temperatures (Fig. 4a,c) is negative, whereas the temporal covariance is positive (Fig. 4e).

Therefore, an important question is whether we can determine the variable driving vegetation changes. This would increase our confidence in the reconstruction. In the simplest case, vegetation patterns across modern space are only determined by the current climate. In this case, the climate variable maximizing the modern spatial correlation, information accessible in the real world, would be the driving variable (Fig. 6a). However, the variable explaining most of the modern spatial vegetation variance was, in our evaluation, not necessarily the one explaining most of the temporal vegetation evolution (compare Figs. 6a vs. 6c).

Therefore, either other parameters beyond modern climate play a role, or the driving variable was not included in our set of six variables. In the model world, and likely in reality, both occur. Evolving parameters such as soil properties partly determine the spatial vegetation distribution, but are constant over time in the model world. On the other hand, the chances of identifying the correct driving variable are also small, as, for example, the length of the growing season might have a stronger influence than summer temperature. What follows from this is that methods that rely only on the modern spatial climate/vegetation

relationship are insufficient to identify the driving variables across time. Here, inverse modeling reconstruction techniques which do not rely on modern spatial calibration sets (Guiot et al., 2009; Yu, 2013) may provide useful additional information. In addition to the downcore tests outlined in Sect. 3.4, a priori expert knowledge on regional ecology is helpful to identify variables of climatic and ecological relevance.

## 4.4 Seasonal bias on reconstructed trends in non-driving variables

In the Northern Hemisphere extratropics of our model world, summer temperature is the variable driving vegetation change across the mid-to-late Holocene. The modern spatial correlation between summer, winter and consequently also mean annual temperatures is positive. Since the modern spatial information determines the downcore temporal reconstruction for all variables, the reconstructions of winter/annual mean temperature changes are biased towards the trend in summer temperatures. What are the implications of such a bias on reconstructions of climate variables which are not primarily influencing vegetation?

Fig. 9 shows the simulated and BMA-reconstructed summer and annual mean temperature for the Northern hemisphere extratropics (all grid boxes north of 50°N). Patterns and magnitudes are highly similar for WA, as well as when only grid boxes with summer/annual mean temperature as dominant variables are picked (not shown). Mid-to-late Holocene summer temperatures are slightly overestimated, but the trend and magnitude are correct. In contrast, the annual mean cooling has the same magnitude as the reconstructed (and simulated) summer cooling – it is exaggerated due to the summer bias in the reconstruction. This could affect the reconstruction of the annual mean temperature evolution of the past 11000 years (Marcott et al., 2013). The reconstructed cooling trend in the mid-late Holocene was stronger than the cooling simulated by climate models. This mismatch is potentially related to a seasonal bias of the reconstruction (Meyer et al., 2015; Liu et al., 2014), and insolation changes as latent and unconsidered variables. Seasonal insolation changes are likely to have direct effects on vegetation by changing the season length, and thus the number of days for growth, and indirect effects by changing local temperatures and their seasonality.

Another example is the comparison between pollen-proxy-based and climate model simulated winter temperature changes between the Last Glacial Maximum and present day, which are stronger in the reconstructions than in the model simulations (Braconnot et al., 2012).

Such a correlation bias on jointly reconstructed climate variables is hard to detect and prove for real-world data. However, the above considerations suggest that for non-driving variables, physically implausible temperature reconstructions may arise due to correlations across modern space. Consequently, estimated temperature trends based on proxy data may appear larger than in the model world, or may have a different shape.

Given our above results, such findings could potentially be explained as changes that are overestimated in the proxy data due to confounding effects of third variables, for example summer length or precipitation changes.

## 4.5 Implications and Outlook

We have focused our analysis on the seasonal evolution of temperatures. However, it is likely that similar biases also affect pollen-assemblage-based reconstructions of other climate variables, such as precipitation. In this light, the result of larger pollen-derived than model simulated precipitation changes between the mid-Holocene and present-day (Braconnot et al., 2012) might be influenced by a reconstruction bias, as the linkage between temperature and precipitation (Trenberth, 2005) may differ across space, time, and timescales (Rehfeld and Laepple, 2016).

Similarly, that modern spatial relationships differ from past temporal relationships might also affect other assemblage-based climate reconstructions. Examples include planktonic foraminifera counts which are used to reconstruct marine temperature changes; in this case, the climate variables include water temperature at different seasons and water depths (Telford et al., 2013). Similar effects might also be in place for other environmental or climate proxies such as chironomids, diatoms and dinoflagellates (Telford and Birks, 2011), which all rely on modern spatial calibration approaches. Consequently, it would be interesting to study ecological, geographical and climatic effects on reconstruction results in other ecological models (e.g. FORAMCLIM, Lombard et al., 2011). In the vegetation model used, the simulated PFTs have broad climatic tolerances (Suppl. Table 1). This might exaggerate the seasonal bias problem, as the winter sensitivity of the simulated vegetation might too be low. While this

would strengthen our general conclusion, that transfer function diagnostics based on modern calibration data alone are not sufficient to characterize reconstructability, it asks for a cautious interpretation of the magnitude of the reconstruction bias.

More work is needed to quantify the impact of seasonality and other secondary variables on temperature estimates based on biomarker proxies, and to develop methods that acknowledge and account for confounding variables in the reconstruction. Repeating this study with a dynamic vegetation model that simulates a larger number of PFTs (Sitch et al., 2003, e.g. LPJ-GUESS), or with models for marine ecology (e.g. FORAMCLIM, Lombard et al., 2011) could provide more insight. Transient paleoclimate model experiments with more complex land surface and biosphere schemes (i.e., with a larger number of PFTs) would be particularly useful to test whether assemblage-based climate reconstruction methods allow for the accurate joint reconstruction of several climate variables.

Future work, extending the conceptual approach in this study, could test

- the reconstructability of multiple climate parameters in an idealized setting. This could be done using artificial vegetation and climate, or a coupled climate model with a vegetation model of higher complexity (than JSBACH) and/or with larger climatic changes. It could also allow in-depth tests for the predictability of reconstruction skill for one or more climate parameters (e.g. using methods described in Juggins, 2013).

- the impact of species richness on the reconstruction error. This could employ vegetation models of different complexity run for the same climate forcing (e.g. by contrasting JSBACH results with LPJ-GUESS results), or random datasets (e.g. Telford and Birks, 2011). Here, it is particularly important to exploit the independence of the modern validation statistics.

- Adding proxy noise and age uncertainty would allow a more in-depth comparison of spatial and temporal errors, and a more representative test of the `randomTF`-algorithm.

A first estimate of potential biases in model-data comparison of multiple climate variables can be obtained through the comparison of simulated spatial and temporal covariances. If they are very different, caution is called for in the interpretation of joint proxy reconstructions of these variables.

## 5   Conclusions

Using a Holocene climate model simulation with interactive vegetation as a testbed, we analyzed the skill and potential biases in pollen-based climate reconstructions. We find that in our model experiments, transfer function reconstruction methods pull the spatial covariances between climate variables through into the downcore temporal reconstructions. As a consequence, temporal changes of a dominant climate variable (for the Northern Hemisphere: often summer temperature) are imprinted on a less important variable (here: often winter temperature), leading to reconstructions biased towards the dominant variable's trends. Given the conceptual nature of our study, we consider these results as primarily illustrative, and have limited ourselves to testing the reconstructability of individual parameters. More work is needed to develop and test methods for the reconstruction of multiple climate parameters and for the predictability of reconstruction skill.

One assumption underpinning transfer-function climate reconstructions is that environmental variables other than those considered in the calibration are not important, or that their relationship with the reconstructed variable(s) is the same in the past as in the modern spatial calibration dataset. In our model world, we have clearly shown that this assumed correlative uniformitarianism is violated, as the modern spatial relationship between climate variables, such as winter and summer temperatures, and the past temporal relationship often differs. Translating this to real world reconstructions would imply that large-scale reconstructions of multiple climate variables need to be carefully considered, as reconstructions of climate variables which are not primarily influencing vegetation can be biased. It would also imply that the driving climate variables cannot be reliably determined by only analyzing the modern spatial climate-vegetation relationship. Therefore, climate variables which actually drove vegetation variability in the past are likely better identified using expert knowledge on ecology, and with statistical analyses involving the fossil vegetation record.

## Appendix A: Acronyms

**PFT** Plant functional type

    **teT** PFT: tropical evergreen trees

    **tdT** PFT: tropical deciduous trees

    **eteT** PFT: extratropical evergreen trees

    **etdT** PFT: extratropical deciduous trees

    **rS** PFT: raingreen shrubs

    **cS** PFT: cold shrubs

    **C3** PFT: C3 grass

    **C4** PFT: C4 grass

    **bS** surrogate PFT: bare soil

**BMA** Best modern analog method (in literature also: Modern Analog approach)

**WA** Weighted averaging

**RDA** Redundancy analysis

**CCA** Canonical correspondence analysis

**RMSE(P)** Root mean square error (of prediction)

**MAT** Mean annual temperature

**MTWA** Mean temperature warmest month

**MTCO**  Mean temperature coldest month

**PANN**  Mean annual precipitation

**MPCO**  Mean precipitation coldest month

**MPWA**  Mean precipitation warmest month

5   **Appendix B: Used software**

All analyses were carried out in the open source environment R, version 3.2.2. Reconstructions were performed using the rioja package (v. 0.9-5), paleosig (v. 1.1-3) and the vegan library (v. 2.3-0). The code is available on request.

*Acknowledgements.*  We gratefully acknowledge Anne Dallmeyer and Johann Jungclaus for providing the ECHAM5/MPIOM model output, Ulrike Herzschuh for discussion and John W. Williams and two anonymous reviewers whose reviews helped us to improve the manuscript.
10   Andrew Dolman is thanked for proof-reading. We thank the Initiative and Networking Fund of the Helmholtz Association (grant VH-NG-900) for funding and the DAAD-PPP program (contract 57160457) for travel support.

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

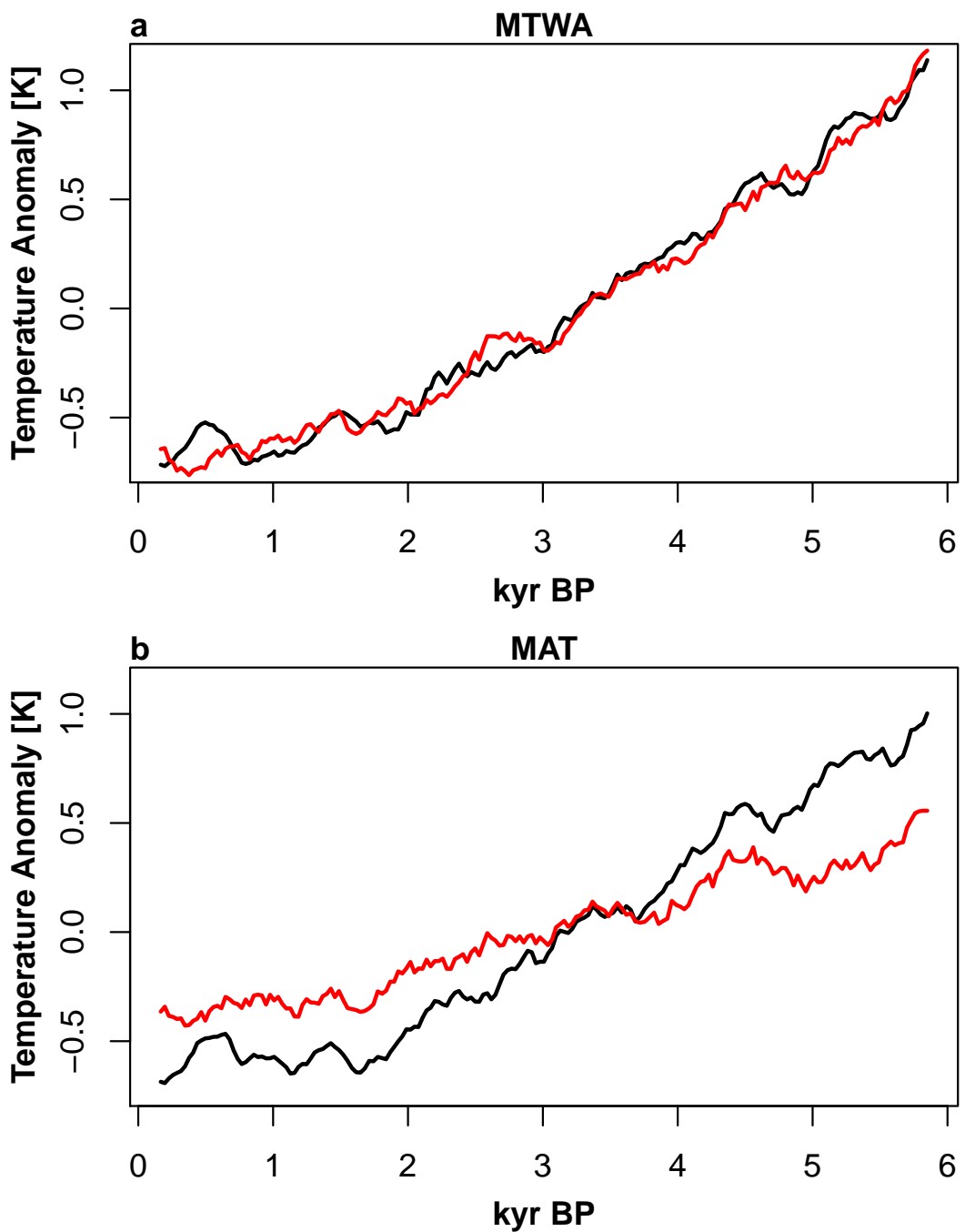

**Figure 9.** Simulated (red) and BMA-reconstructed (black) extratropical mean temperature changes over the 6k-run (BMA). The amplitude of the summer temperature trends (a) agree well, whereas the amplitude for the simulated mean annual temperature change (b) is overestimated in the reconstructions.