# Peer review of "Assessing performance and seasonal bias of pollen-based climate reconstructions in a perfect model world"

_Climate of the Past, 2016_

## Referee Comment (RC1) · Anonymous Referee #1 · 24 Feb 2016

Rehfeld et al. Assessing performance and seasonal bias of pollen-based climate reconstructions in a perfect model world

The authors use a vegetation model and climate model to simulate the process of reconstructing climate from pollen data, and in turn to assess the ability of pollen-based methods to accurately reconstruct seasonal Holocene climate change.

This is a interesting and novel approach, and although similar virtual experiments have been conducted with other proxies, this is the first time that I know of where it has been applied to pollen. Pollen-based climate reconstructions have been widely used in data-model comparisons, and large discrepancies have been found between these reconstructions and climate model simulations during the Holocene, particularly in terms

of seasonality. Investigation of potential seasonal bias in pollen-based reconstructions is therefore of particular interest and importance.

The study is generally well written and presented, but has a number of critical issues that I do not think can be easily resolved. The most obvious of these is the unrealistically low number of virtual taxa, or in this case PFTs, used in the transfer-function. To some extent the authors themselves acknowledge this (lines 550-554) "Furthermore, the methods we have tested are limited by the low number of plant functional types, as large-scale PFT-based pollen reconstructions use roughly 2-3 times the number of PFTs (as e.g. in Davis et al., 2003; Mauri et al., 2014)." This is actually an underestimate, since of the 8 PFT's used by Rehfeld et al, 3 are tropical (tropical evergreen trees, tropical deciduous trees and C4 grasses), leaving just 5 PFT's for the extra-tropics such as Europe. Davis et al. 2003 and Mauri et al. 2014 use 22 PFT's for Europe, which is more than 4 times the number used by the authors in their study. Taxa based pollen-climate transfer functions commonly use upwards of 50-60 taxa. These numbers are important because the individual behavior of the PFTs/Taxa and their climatic tolerances constitute the degrees of freedom necessary to reconstruct multiple climatic variables, and particularly those that may show close co-variance as cited as a potential problem by the authors. Furthermore, the PFT's used in the study by Rehfeld et al. have extremely broad climatic tolerances (deciduous trees, evergreen trees, grass..) that can be expected to have little diagnostic power. No pollen-climate transfer function should or would be based on such a low number of taxa/PFT's with such broad climatic sensitivity, and it is therefore disingenuous of the authors to compare their own over-simplified approach with the approach used in actual pollen-climate reconstructions. For instance the authors infer that because they were unable to reliably reconstruct winter temperatures, this should also be a problem for actual pollen-climate reconstructions. In reality, the problem with winter temperatures is just as likely to be a result of the authors over-simplified experimental design and the use of a limited number of PFT's with limited winter temperature sensitivity. This problem is likely to be compounded by the use of climate data for calibration from a climate model with

low spatial resolution, and where the spatial variability of climate is highly smoothed compared to the real world. On the one hand this reduces the variance of climate and vegetation in the training set and on the other, it greatly increases the propensity for spatial auto-correlation that the authors also highlight as a problem in their study. Whilst some simplification should be expected in a 'virtual' study like this, it is important not to over-simplify to the point where the study itself is so far removed from any actual application that the results are not comparable. The problem here is that the authors consistently conflate their results with those from actual pollen-climate reconstructions (as in the title), and therefore are at risk of presenting a fallacious argument that the average reader who is not so familiar with the topic will likely interpret at face value.

The subject of the paper is nevertheless interesting, and one that would otherwise be worthy of publication. I would therefore encourage the authors to collaborate with someone who has more experience in pollen-climate modeling, and to use a vegetation model such as LPJGUESS which can simulate a greater number of PFT's/Taxa so that the analysis can be more comparable with how pollen-climate transfer functions are actually applied.

---

## Referee Comment (RC2) · Anonymous Referee #2 · 29 Feb 2016

The paper by Rehfeld et al. deals with the pollen-based climate reconstructions. The authors use climate model data and modelled vegetation to explore the reliability of reconstructions of different climate parameters in pollen-based reconstructions. The advantage in such an experiment "in an ideal model world" is that the past climate and vegetation are known at all times (6 ka to present), allowing to assess the reliability of the reconstructions. The authors show that reconstructing multiple climate parameters can be misleading, as it is possible that in reality there is only one climate parameter which drives the spatial and temporal vegetation change, and the reconstructions of other climate parameters show temporal variability which is caused by the fact that these less important parameters are spatially correlated with the important parameter
in the modern spatial data used for constructing the transfer function. This is certainly nothing new, most of the palaeoecologists using pollen data have been aware of this problem, but it is useful to have a special study where this problem in explicitly explored using novel approaches.

I find it easy to agree with the authors that "the temporal changes of a dominant climate variable are imprinted on a less important variable, leading to reconstructions biased towards the dominant variable's trend" and that the high r2 in the cross-validation is of limited use to identify which variables can be reconstructed, as r2 can be high not only for the variable which is really important for vegetation or pollen, but also to non-important variable which are spatially correlated with the important variable. The authors suggest assessing the amount of fossil vegetation variance explained the reconstruction output and expert knowledge as possible means to select the climate variables. The latter one has been used in pollen-based reconstructions, but unfortunately the expert knowledge almost invariably is limited to present ecological setting. It is possible, or even likely, that if we go back in time enough, the combination of climate parameters governing the vegetation composition have been fundamentally different from the present.

There is one striking problem with the paper. Given that the authors use model data only, they are restricted to use plant functional types (pft), not pollen types or plant species. In the real world, the WA-based climate reconstructions often comprise over 100 pollen types, not pfts. Modern analogue-based reconstructions use pfts, but even in them the number of pfts is generally 20-30. In a striking contrast, the number of pfts in the current study is eight - in other words extremely low. I am surprised that the palaeoclimate reconstructions with such a low number of variables make any sense in the first place, given that they are based on a few, extremely broad pft classes. I suspect that the reconstructions using modern analogue must have included some serious problems which are not reported in the paper. The problem of multiple analogues (where the many modern analogues for the fossil sample are present, often in very dif-

ferent climatic settings) would be unavoidable with eight pdfs only. The error estimates of the calibration sets and the fossil reconstructions are not presented or discussed in the paper, but they most likely are extremely high. I therefore wonder if the difference in the reconstructions (in Fig. 8 and 9, for example) are inside or outside the error estimates?

———————————————

---

## Author Comment (AC1) · 4 Apr 2016

**Reply to anonymous reviewer comments, 24.2.2014, concerning the manuscript of Rehfeld, Trachsel, Telford & Laepple in discussion for Climate of the Past (doi:10.5194/cp-2016-13)**

04/04/16

**Summary**

We would like to thank the reviewer for his/her detailed comments which will help to improve the clarity and quality of the manuscript.

The reviewer mainly comments on limitations of our study related to the coarse representation of the simulated vegetation and the simulated climate. These are valid points which we are aware of. We will ensure that they are more properly represented in the revised manuscript. However, as we demonstrate below, the reviewer's points are overstated and the limitations brought up by the reviewer do not affect our main conclusions.

Reviewer's comments are given in grey. Emphasis in *italics* was added to highlight main points.

**Point 1: Number of taxa used in the study**

> **Reviewer's comments:**
> The authors use a vegetation model and climate model to simulate the process of reconstructing climate from pollen data, and in turn to assess the ability of pollen-based methods to accurately reconstruct seasonal Holocene climate change.
> This is a interesting and novel approach, and although similar virtual experiments have been conducted with other proxies, this is the first time that I know of where it has been applied to pollen. Pollen-based climate reconstructions have been widely used in data-model comparisons, and large discrepancies have been found between these reconstructions and climate model simulations during the Holocene, particularly in terms of seasonality. Investigation of potential seasonal bias in pollen-based reconstructions is therefore of particular interest and importance.
> *The study is generally well written and presented, but has a number of critical issues that I do not think can be easily resolved. The most obvious of these is the unrealistically low number of virtual taxa, or in this case PFTs, used in the transfer-function.* To some extent the authors themselves acknowledge this (lines 550-554) "Furthermore, the methods we have tested are limited by the low number of plant functional types, as large-scale PFT-based pollen reconstructions use roughly 2-3 times the number of PFTs (as e.g. in Davis et al., 2003; Mauri et al., 2014)." *This is actually an underestimate, since of the 8 PFT's used by Rehfeld et al, 3 are tropical (tropical evergreen trees, tropical deciduous trees and C4 grasses), leaving just 5 PFT's for the extratropics such as Europe.* Davis et al. 2003 and Mauri et al. 2014 use 22 PFT's for Europe, which is more than 4 times the number used by the authors in their study. *Taxa based pollen-climate transfer functions commonly use upwards of 50-60 taxa.* These numbers are important because the individual behavior of the PFTs/Taxa and their climatic tolerances constitute the degrees of freedom necessary to reconstruct multiple climatic variables, and particularly those that may show close covariance as cited as a potential problem by the authors.

As we state in the manuscript, the number of  Plant Functional Types (PFTs) in our model study is lower than what is generally used in a real-world large-scale reconstruction exercise (eg. in Davis et al., 2003, Mauri et al., 2014, Mauri et al., 2015). We use 9 PFTs (one of which is representing bare soil, or desert fraction), whereas e.g. Mauri et al. (2014) use 22 (two of which are virtual). In a given region, the number of contributing PFTs is lower, as some PFTs only appear in some region, thus leading the reviewer to conclude that the difference between our modeled and the real world PFTs is up to factor of four.

However, what is ultimately relevant for the calibration and reconstruction efforts is the information contributed by the PFTs or taxa; in other words how many of them actually contribute to the pollen (or PFT) diagram in a relevant way. This strongly differs from the number of PFTs/ the number of taxa.

The effective number of species can be quantified by the Hill's number N2, which is an entropy-based measure for the vegetation diversity (Hill, 1973). An  analysis of  4990 sites in the  European modern pollen database, which formed the basis of Mauri et al. (2014, 2015) and many other large-scale pollen-based reconstructions shows that the median effective number of species (N2) is 4.9, much lower than the median palynological richness of 31 taxa  (Fig R1 below) . If the pollen data of the European Pollen Database were assigned to PFTs we would expect the N2 for the PFTs to be even lower.

In our study we only use sites (grid points) with a Hill's number larger than 2 (p.4, l. 19). Of the 458 grid points we use, the median N2 for the fossil data (shown in Fig. 1d in the manuscript) is 2.9, and for the modern calibration data it is 2.7. Therefore, although the number of PFTs is much lower in the model, the diversity and effective number of species is not much lower than that in actual pollen-climate reconstructions.

In the revised version of the manuscript, we will discuss the differing number of taxa as well as the difference in the effective vegetation diversity.

*Figure R1: Distribution of the number of species (left) and effective number of species (N2) in modern samples in the European Pollen Database.*

[Figure]

[Figure]

**Point 2: Winter vs. summer temperature reconstructability**

> Furthermore, the PFT's used in the study by Rehfeld et al. have extremely broad climatic tolerances (deciduous trees, evergreen trees, grass..) that can be expected to have little diagnostic power. No pollen-climate transfer function should or would be based on such a low number of taxa/PFT's with such broad climatic sensitivity, and it is therefore disingenuous of the authors to compare their own over-simplified approach with the approach used in actual pollen-climate reconstructions. For instance the authors infer that because they were unable to reliably reconstruct winter temperatures, this should also be a problem for actual pollen-climate reconstructions. In reality, the problem with winter temperatures is just as likely to be a result of the authors over-simplified experimental design and the use of a limited number of PFT's with limited winter temperature sensitivity.

In our study we have followed the standard workflow for pollen-based reconstructions. The reconstructability, or non-reconstructability, of climate variables is often inferred from the transfer function r2 and the RMSEP in cross-validation (e.g. in Mauri et al., 2014, Frechette et al., 2008). These test diagnostics are based on the modern calibration data alone. As we show in Fig. 7 in the manuscript, the transfer function estimated r2 for winter temperatures is similar to that for other temperature variables. Therefore, the transfer function diagnostics suggest, that winter temperature is reconstructible. We agree with the reviewer that the true reason for low *actual* reconstructability of winter temperatures may well be that winter temperatures have little influence on the modeled vegetation (which might be realistic in at least in some regions of the world, such as Siberia).

However, even if the winter sensitivity of the model vegetation were unrealistically low, this would only strengthen our conclusion that transfer function diagnostics based on modern calibration data alone are not sufficient to characterize reconstructability.

**Point 3: Resolution of the calibration climate dataset**

> This problem is likely to be compounded by the use of climate data for calibration from a climate model with low spatial resolution, and where the spatial variability of climate is highly smoothed compared to the real world. On the one hand this reduces the variance of climate and vegetation in the training set and on the other, it greatly increases the propensity for spatial auto-correlation that the authors also highlight as a problem in their study.

We of course agree that the spatial climate fields from our climate model simulations have a much lower resolution than for example the 0.5 minute resolution interpolated instrumental dataset (Hijmans et al. 2005) used in Mauri et al. (2015).
However, the resolution in itself is neither the determining factor for the variance of the climate explanatory variable, nor for the spatial autocorrelation. Most importantly, the covariance structure between the different climate variables (e.g. summer and winter temperatures), is not directly a function of the resolution. Given the large-scale structure of spatial climate and especially temperature variations (e.g. Hansen, & Lebedeff, 1987) we do not expect a strong influence of the resolution, except in some areas where elevational gradients, not well represented in the coarse model topography.

This is demonstrated in the Figure R2 below, which compares the distribution and relation of the gridpoint winter and summer temperatures in the 0.5 minute resolution temperature field (Hijmans et al. 2005) with the gridpoint temperatures from our low resolution ECHAM5-MPIOM simulation in Europe (30-60N, 0-30E) including several mountain ranges. The 61 land grid-points from the climate model cover most of the phase-space spanned by the 13 million grid points of the 0.5min resolution field, except high-altitude regions represented by the lower-left tail. The model field further shows a similar correlation between the seasons. Thus we see no reason to expect that the low resolution would bias our results towards less skill in reconstructing multiple variables. We will include a detailed discussion in the revised manuscript.

[Figure]

*Figure R2: Distribution and relation of gridpoint winter and summer temperatures in the 0.5'-resolution temperature field of (Hijmans et al., 2005) and the gridpoint temperatures of the ECHAM5-MPIOM simulation (Fischer & Jungclaus, 2011) used in this study.*

Spatial autocorrelation is often not considered in papers reconstructing climate from pollen (e.g. in Bartlein et al., 2011), and will tend to be a larger problem in the densely sampled pollen databases than in our low resolution data, as each pollen site has many geographically close neighbors which can be used as an analogue in the modern analogue technique.

**Point 4: Simplification**

*Whilst some simplification should be expected in a 'virtual' study like this, it is important not to over-simplify to the point where the study itself is so far removed from any actual application that the results are not comparable.* The problem here is that the authors consistently conflate their results with those from actual pollen-climate reconstructions (as in the title), and therefore are at risk of presenting a fallacious argument that the average reader who is not so familiar with the topic will likely interpret at face value.

We fully agree that the complexity of the vegetation representation in the model as well as the simulated climate evolution are a strong simplification of the reality. Therefore, results on the Holocene evolution of specific PFT's, the actual spatial pattern of PFT's, or the reconstructability of a certain climate variable in a certain region should not be directly translated to the real-world.

On the other hand, conclusions about reconstruction methods and the relation of spatial calibration and downcore reconstruction only require a consistent dataset of climate and vegetation parameters in space and time and do not depend on details of the climate evolution or vegetation response, as long as the dataset is realistic enough that we can apply the real world reconstruction workflow. The major factor shaping these results is that the modern spatial relationships between climate variables is different from the changes in the relationships over time, which is a robust feature related to the transient insolation forcing.

In the revision, we will check in detail again if all our statements are either independent from the model-world specifics , or are clearly marked that they just apply to the model world. Furthermore, we will emphasize the limitations of our study further by extending paragraph 4.1, and by highlighting their impact in the conclusion paragraphs. We do, however, find the title of our manuscript appropriate, as it clearly expresses that we work with pollen-related methods in an ideal model world.

**Point 5: Repeating the study with a more elaborate vegetation model (LPJGuess)**

The subject of the paper is nevertheless interesting, and one that would otherwise be worthy of publication. I would therefore encourage the authors to collaborate with someone who has more experience in pollen-climate modeling, and to use a vegetation model such as LPJGUESS which can simulate a greater number of PFT's/Taxa so that the analysis can be more comparable with how pollen-climate transfer functions are actually applied.

As we have shown above, our analysis is comparable to actual pollen-climate transfer functions, as the effective number of species present in the model data is within the range of the numbers observed for the European Pollen Database, which formed the basis e.g. for (Mauri et al., 2014; Mauri et al., 2015).

We agree that using a more complex vegetation model, such as LPJGUESS, would be worthwhile in a future study as this would not only increase the number of PFT's but also include a more realistic interaction between climate and vegetation. This would allow to also test other properies of the reconstruction e.g. effects on the time-scale dependent variability, or potential time-lags. We will thus include this proposal in the outlook of our study.

However, we expect that the main result of this manuscript, the limitations of spatial modern calibrations, would only be strengthened. This is because a more realistic climate-vegetation response will even differ more in time versus space, if for example the modern vegetation is not yet in equilibrium with the modern climate state.

---

## Author Comment (AC2) · 15 Apr 2016

**Reply to anonymous reviewer comments, 29.2.2014, concerning the manuscript of Rehfeld, Trachsel, Telford & Laepple in discussion for Climate of the Past (doi:10.5194/cp-2016-13)**

04/15/16

**Summary**

We would like to thank the reviewer for her/his insightful remarks, which will help us to improve a revised manuscript.

Reviewer's comments are given in grey. Emphasis in *italics* was added to highlight main points.

**Point 1: Combination of climate parameters governing vegetation composition in the past**

The paper by Rehfeld et al. deals with the pollen-based climate reconstructions. The authors use climate model data and modelled vegetation to explore the reliability of reconstructions of different climate parameters in pollen-based reconstructions. The advantage in such an experiment "in an ideal model world" is that the past climate and vegetation are known at all times (6 ka to present), allowing to assess the reliability of the reconstructions. *The authors show that reconstructing multiple climate parameters can be misleading, as it is possible that in reality there is only one climate parameter which drives the spatial and temporal vegetation change, and the reconstructions of other climate parameters show temporal variability which is caused by the fact that these less important parameters are spatially correlated with the important parameter in the modern spatial data used for constructing the transfer function.* This is certainly nothing new, most of the palaeoecologists using pollen data have been aware of this problem, but it is useful to have a special study where this problem in explicitly explored using novel approaches.

I find it easy to agree with the authors that "the temporal changes of a dominant climate variable are imprinted on a less important variable, leading to reconstructions biased towards the dominant variable's trend" and that the high r2 in the cross-validation is of limited use to identify which variables can be reconstructed, as r2 can be high not only for the variable which is really important for vegetation or pollen, but also to non-important variable which are spatially correlated with the important variable. The authors suggest assessing the amount of fossil vegetation variance explained the reconstruction output and expert knowledge as possible means to select the climate variables. *The latter one has been used in pollen-based reconstructions, but unfortunately the expert knowledge almost invariably is limited to present ecological setting.* It is possible, or even likely, that *if we go back in time enough, the combination of climate parameters governing the vegetation composition have been fundamentally different from the present.*

Non-analogue combinations of climate and other physiologically relevant variables certainly occurred in the past and will result in less accurate reconstructions. For example, the effect of low-CO2 concentrations on LGM pollen-based reconstructions is difficult to quantify and is hence largely ignored. Climatic conditions in the Holocene, the period analysed in this study are unlikely to have been sufficiently non-analogous to cause serious problems, and non-equilibrium vegetation may be more of a general problem.

**Point 2: Number of PFTs**

There is one striking problem with the paper. Given that the authors use model data only, they are restricted to use plant functional types (pft), not pollen types or plant species. In the real world, the WA-based climate reconstructions often comprise over 100 pollen types, not pfts. Modern analogue-based reconstructions use pfts, but even in them the number of pfts is generally 20-30. In a striking contrast, the number of pfts in the current study is eight - in other words extremely low. I am surprised that the palaoeclimate reconstructions with such a low number of variables make any sense in the first place, given that they are based on a few, extremely broad pft classes.

We thank the reviewer for raising this point. We agree that the number of Plant Functional Types (PFTs) in our model study is lower than what is generally used in a real-world large-scale reconstruction exercise (eg. in Davis et al., 2003, Mauri et al., 2014, Mauri et al., 2015). We use 9 PFTs (one of which is representing bare soil, or desert fraction), whereas e.g. Mauri et al. (2014) use 22 (two of which are virtual). However, as pointed out in the response to Reviewer 1, what is ultimately relevant for the calibration and reconstruction efforts is the information contributed by the PFTs or taxa; in other words how many of them actually contribute to the pollen (or PFT) diagram in a relevant way. This number is much lower than the number of PFTs, or the number of taxa in a pollen diagram, but as we outline below, this number is comparable between our modeled PFTs and real-world pollen diagrams.

The median palynological richness in the 4990 sites in the European Pollen Database is 31. However, the median effective number of species (given by Hill's N2 (Hill, 1973)), discounting very rare taxa which would not be relevant for reconstructions, is 4.9. Of the 458 grid points we use, the median N2 for the fossil data (shown in Fig. 1d in the manuscript) is 2.9, and for the modern calibration data it is 2.7. Therefore, although the number of PFTs is much

[Figure]

*Figure R1: Species response curves for the calibration radius around example site 120E,72N. Only non-zero taxa are shown.*

[Figure]

*Figure R2: Ratio of the standard deviation of the MTWA climate variable at modern analog sites over the standard deviation within the training set.*

lower in the model, the diversity and effective number of species is not much lower than that many actual pollen-climate reconstructions. Moreover, we are setting bounds on N2 and turnover to avoid pathological problems due to too few taxa. This is an important aspect we will discuss in more detail than before in a revised manuscript.

**Point 3: Multiple Analogues**

*I suspect that the reconstructions using modern analogue must have included some serious problems which are not reported in the paper. The problem of multiple analogues (where the many modern analogues for the fossil sample are present, often in very different climatic settings) would be unavoidable with eight pdfs only.*

We thank the reviewer for raising this important point. The multiple analog problem could arise if the species response curve (e.g. with respect to the climate variable, e.g. MTWA) within a modern calibration radius was multimodal. However, analyzing the species response curves at several sites suggests that this is not the case. As an example, Fig. R3 shows the species response curves for all taxa effectively present in the Siberian site for which we also show the complete reconstruction workflow in the manuscript (Fig. 2). The species response curves are not multimodal.

Furthermore, the overall high transfer function r2 (Fig. 7 in the manuscript) shows, that analogs are not picked at random from the training set, and underlines that multiple analogs are not a problem . To exemplify this we calculate the ratio of the standard deviations of the temperatures at the analog sites, and the standard deviation of the temperatures across the whole training sets (Fig. R4) . The ratios are generally smaller than 0.5, thus illustrating that the analog sites are not randomly drawn from the training set.

In the revised version of the manuscript we will explicitly demonstrate that arbitrary analogs are not a problem here, by including the above discussion.

**Point 4: Error estimates and uncertainties**

*The error estimates of the calibration sets and the fossil reconstructions are not presented or discussed in the paper, but they most likely are extremely high. I therefore wonder if the difference in the reconstructions (in Fig. 8 and 9, for example) are inside or outside the error*

*estimates*?

For the sake of brevity we have not given explicit plots of all quality metrics within the manuscript. We do, however, give standard error estimates (RMSEP, cross-validation r2) for the example grid point reconstruction workflow (Fig. 2 in the manuscript), and for all grid points >50N (Table 1). The calibration r2 for all climate variables is given in Fig. 7 in the manuscript, and Fig. 7 in the supplement compares the different error estimates for downcore RMSE and RMSEP for warmest month temperature. While we have given the summary numbers for the RMSEP and the r2 in Table 1, we agree that it would be helpful to show these statistics explicitly for each gridpoint, projected onto a map. We show the results for the temperature variables below in Fig. R5. As suspected by the reviewer, the RMSEP, particularly for MTCO, is not small, however, it is not much larger than that in real-world large-scale reconstructions (c.p. Frechette et al., 2008; Mauri et al., 2014). The regions with high RMSEP are largely consistent with the regions where the calibrations' r2 is low (Fig. 7 in the manuscript). We will improve Figs. 4 and 5 in the manuscript by masking grid points with an r2 below 0.5, as the results show the covariance across time and space of winter and summer temperature reconstructions, and could be affected by this. Furthermore we plan to incorporate Fig. R5 in the supplement of the revised manuscript, and we will also better explain and more prominently discuss Table 1 in the revised manuscript.

[Figure]

*Figure R3: RMSEP for 10-fold-leave-group-out cross validation of MTCO (a), MTWA (b) and MAT (c). Black gridcells indicate a RMSEP larger than 5C.*

The question whether or not the temperature changes against time shown in Fig. 8 and 9 in the manuscript are significant or not using the calibration RMSEP is not straightforward. A standard assumption in paleoclimate reconstructions is that errors in time and space are independent (as assumed e.g. in Marcott et al. 2013, Fedorov et al., 2013, Shakun et al., 2012). This assumption would result in a standard error of 0.008C (1se) for the difference of -0.72C between the calibration at 0k vs. 6k in Fig. 8 (taken across all 27 gridpoints >70N and 197 time points). For Fig. 9 the difference at the 0k is 0.32C, at 6k it is .44C and the standard error at these time points is smaller than 0.13C. Both differences in the reconstructions would therefore be highly significant. On the other hand, there are good reasons to expect spatial and temporal correlations in the errors. In the (unrealistic) extreme case of a complete dependency or errors, the differences would be not significant. In reality the true uncertainty likely lies between the two extremes assumed here but a mechanistic understanding of processes causing the proxy uncertainty is required to provide better error estimates. In the revised manuscript, will discuss these aspects of uncertainty in Section 4 and include the uncertainty estimates for both cases.

---

## Referee Report (RR1)

**Author response to reviewer and editor comments**

**Manuscript: "Assessing performance and seasonal bias of pollen-based climate reconstructions in a perfect model world" by K. Rehfeld, M. Trachsel, R.J. Telford and T. Laepple (doi:10.5194/cp-2016-13)**

**05/17/16**

**Response to editor comments**

The main criticism, expressed by both reviewers, concerning our work was that the number of Plant Functional Types in our model world is much lower (9) than in comparable real-world reconstructions based on PFTs (22, e.g. Mauri et al., 2014), or on pollen directly (where sometimes more than 4 times as many taxa are distinguished). In our response we argue that, for the purpose of climate reconstructions, the number of taxa present in relevant proportions is more important than the overall number of taxa distinguished, and that our efficient number of taxa was not unusually low. We illustrated this by showing the difference between the median effective number of species (4.9), and the median palynological richness (31) in the European modern Pollen database. We added a new paragraph to the main text and included a new supplementary figure (SFig. 1) to reflect these findings.

The editor suggested that this point could be further discussed and extended as it might imply that field work and reconstructions could be done more effectively. We performed own preliminary analysis of European Holocene pollen stratigraphies, which confirm that reconstructions obtained from greatly reduced datasets are highly similar to full reconstructions. The picture may, however, be different in different climatological and ecological settings, e.g. in the Tropics. Furthermore, the presence of rare taxa may play a very important role in paleoecological studies, which, to many researchers performing the field studies, are more relevant than paleoclimate reconstructions. Therefore, we would like to refrain from extending this disucssion in the manuscript, as it would deviate from our key point, the different relationships across space and time in vegetation and climate and their effects on the reconstructions. Having said this, we agree with the editor that this is an interesting topic for a further study.

**Summary of changes**

In the revised version of the manuscript we have addressed all points made by the reviewers. We have added several new paragraphs and four new figures in the supplementary material. We improved the linkage between the main text and the figures and tables in the main text and the supplementary. We furthermore revised two figures, masking out grid points which showed a low transfer function performance, and rephrased statements throughout the manuscript. Finally, we corrected initials in several citations.

We give our previous response to the two anonymous reviews below. At the bottom of each key point made by the reviewers, and below our response, we outline in **dark green** what changes we have made in the manuscript. Line and page numbers refer to the attached document with highlighted changes.

**Response to anonymous reviewer #1**

**Summary**

We would like to thank the reviewer for his/her detailed comments which will help to improve the clarity and quality of the manuscript.

The reviewer mainly comments on limitations of our study related to the coarse representation of the simulated vegetation and the simulated climate. These are valid points which we are aware of. We will ensure that they are more properly represented in the revised manuscript. However, as we demonstrate below, the reviewer's points are overstated and the limitations brought up by the reviewer do not affect our main conclusions.

Reviewer's comments are given in grey. Emphasis in *italics* was added to highlight main points.

**Point 1: Number of taxa used in the study**

> **Reviewer's comments:**
> The authors use a vegetation model and climate model to simulate the process of reconstructing climate from pollen data, and in turn to assess the ability of pollen-based methods to accurately reconstruct seasonal Holocene climate change.
> This is a interesting and novel approach, and although similar virtual experiments have been conducted with other proxies, this is the first time that I know of where it has been applied to pollen. Pollen-based climate reconstructions have been widely used in data-

model comparisons, and large discrepancies have been found between these reconstructions and climate model simulations during the Holocene, particularly in terms of seasonality. Investigation of potential seasonal bias in pollen-based reconstructions is therefore of particular interest and importance.

*The study is generally well written and presented, but has a number of critical issues that I do not think can be easily resolved. The most obvious of these is the unrealistically low number of virtual taxa, or in this case PFTs, used in the transfer-function.* To some extent the authors themselves acknowledge this (lines 550-554) "Furthermore, the methods we have tested are limited by the low number of plant functional types, as large-scale PFT-based pollen reconstructions use roughly 2-3 times the number of PFTs (as e.g. in Davis et al., 2003; Mauri et al., 2014)." *This is actually an underestimate, since of the 8 PFT's used by Rehfeld et al, 3 are tropical (tropical evergreen trees, tropical deciduous trees and C4 grasses), leaving just 5 PFT's for the extratropics such as Europe.* Davis et al. 2003 and Mauri et al. 2014 use 22 PFT's for Europe, which is more than 4 times the number used by the authors in their study. *Taxa based pollen-climate transfer functions commonly use upwards of 50-60 taxa.* These numbers are important because the individual behavior of the PFTs/Taxa and their climatic tolerances constitute the degrees of freedom necessary to reconstruct multiple climatic variables, and particularly those that may show close co-variance as cited as a potential problem by the authors.

As we state in the manuscript, the number of  Plant Functional Types (PFTs) in our model study is lower than what is generally used in a real-world large-scale reconstruction exercise (eg. in Davis et al., 2003, Mauri et al., 2014, Mauri et al., 2015). We use 9 PFTs (one of which is representing bare soil, or desert fraction), whereas e.g. Mauri et al. (2014) use 22 (two of which are virtual). In a given region, the number of contributing PFTs is lower, as some PFTs only appear in some region, thus leading the reviewer to conclude that the difference between our modeled and the real world PFTs is up to factor of four.

However, what is ultimately relevant for the calibration and reconstruction efforts is the information contributed by the PFTs or taxa; in other words how many of them actually contribute to the pollen (or PFT) diagram in a relevant way. This strongly differs from the number of PFTs/ the number of taxa.

The effective number of species can be quantified by the Hill's number N2, which is an entropy-based measure for the vegetation diversity (Hill, 1973). An  analysis of  4990 sites in the  European modern pollen database, which formed the basis of Mauri et al. (2014, 2015) and many other large-scale pollen-based reconstructions shows that the median effective number of species (N2) is 4.9, much lower than the median palynological richness of 31 taxa  (Fig R1 below) . If the pollen data of the European Pollen Database were assigned to PFTs we would expect the N2 for the PFTs to be even lower.

In our study we only use sites (grid points) with a Hill's number larger than 2 (p.4, l. 19). Of the 458 grid points we use, the median N2 for the fossil data (shown in Fig. 1d in the manuscript) is 2.9, and for the modern calibration data it is 2.7. Therefore, although the number of PFTs is much lower in the model, the diversity and effective number of species is not much lower than that in actual pollen-climate reconstructions.

In the revised version of the manuscript, we will discuss the differing number of taxa as well as the difference in the effective vegetation diversity.

*Figure R1: Distribution of the number of species (left) and effective number of species (N2) in modern samples in the European Pollen Database.*

[Figure]

[Figure]

**Implemented changes in the revised manuscript:**

**- inserted new discussion and definition of Hill's N2 on p. 5 l. 19.**

**- main text paragraph p.5 l. 30 – p. 6. l. 10 on N2 vs. richness**

**- new  Supplementary Figure SFig. 2**

**Point 2: Winter vs. summer temperature reconstructability**

> Furthermore, the PFT's used in the study by Rehfeld et al. have extremely broad climatic tolerances (deciduous trees, evergreen trees, grass..) that can be expected to have little diagnostic power. No pollen-climate transfer function should or would be based on such a low number of taxa/PFT's with such broad climatic sensitivity, and it is therefore disingenuous of the authors to compare their own over-simplified approach with the approach used in actual pollen-climate reconstructions. For instance the authors infer that because they were unable to reliably reconstruct winter temperatures, this should also be a problem for actual pollen-climate reconstructions. In reality, the problem with winter temperatures is just as likely to be a result of the authors over-simplified experimental design and the use of a limited number of PFT's with limited winter temperature sensitivity.

In our study we have followed the standard workflow for pollen-based reconstructions. The reconstructability, or non-reconstructability, of climate variables is often inferred from the transfer function r2 and the RMSEP in cross-validation (e.g. in Mauri et al., 2014, Frechette et al., 2008). These

test diagnostics are based on the modern calibration data alone. As we show in Fig. 7 in the manuscript, the transfer function estimated r2 for winter temperatures is similar to that for other temperature variables. Therefore, the transfer function diagnostics suggest, that winter temperature is reconstructible. We agree with the reviewer that the true reason for low *actual* reconstructability of winter temperatures may well be that winter temperatures have little influence on the modeled vegetation (which might be realistic in at least in some regions of the world, such as Siberia). However, even if the winter sensitivity of the model vegetation were unrealistically low, this would only strengthen our conclusion that transfer function diagnostics based on modern calibration data alone are not sufficient to characterize reconstructability.

**Implemented changes in the revised manuscript:**

- **new statement on resolution (p. 4 l. 6 – 7)**

- **masking of grid points with low transfer function performance in Figs. 4 & 5**

- **discussion of high RMSEP for winter temperature (p. 10 l. 7-10)**

- **new  Supplementary Figure SFig. 1**

**Point 3: Resolution of the calibration climate dataset**

> This problem is likely to be compounded by the use of climate data for calibration from a climate model with low spatial resolution, and where the spatial variability of climate is highly smoothed compared to the real world. On the one hand this reduces the variance of climate and vegetation in the training set and on the other, it greatly increases the propensity for spatial auto-correlation that the authors also highlight as a problem in their study.

We of course agree that the spatial climate fields from our climate model simulations have a much lower resolution than for example the 0.5 minute resolution interpolated instrumental dataset (Hijmans et al. 2005) used in Mauri et al. (2015).
However, the resolution in itself is neither the determining factor for the variance of the climate explanatory variable, nor for the spatial autocorrelation. Most importantly, the covariance structure between the different climate variables (e.g. summer and winter temperatures), is not directly a function of the resolution. Given the large-scale structure of spatial climate and especially temperature variations (e.g. Hansen, & Lebedeff, 1987) we do not expect a strong influence of the resolution, except in some areas where elevational gradients, not well represented in the coarse model topography.

This is demonstrated in the Figure R2 below, which compares the distribution and relation of the gridpoint winter and summer temperatures in the 0.5 minute resolution temperature field (Hijmans et al. 2005) with the gridpoint temperatures from our low resolution ECHAM5-MPIOM simulation in Europe (30-60N, 0-30E) including several mountain ranges. The 61 land grid-points from the climate model cover most of the phase-space spanned by the 13 million grid points of the 0.5min resolution field, except high-altitude regions represented by the lower-left tail. The model field further shows a

similar correlation between the seasons. Thus we see no reason to expect that the low resolution would bias our results towards less skill in reconstructing multiple variables. We will include a detailed discussion in the revised manuscript.

[Figure]

*Figure R2: Distribution and relation of gridpoint winter and summer temperatures in the 0.5'-resolution temperature field of (Hijmans et al., 2005) and the gridpoint temperatures of the ECHAM5-MPIOM simulation (Fischer & Jungclaus, 2011) used in this study.*

Spatial autocorrelation is often not considered in papers reconstructing climate from pollen (e.g. in Bartlein et al., 2011), and will tend to be a larger problem in the densely sampled pollen databases than in our low resolution data, as each pollen site has many geographically close neighbors which can be used as an analogue in the modern analogue technique.

**Implemented changes in the revised manuscript:**

- **new statement on resolution (p. 4 l. 6 – 7)**

- **new  Supplementary Figure SFig. 1**

**Point 4: Simplification**

> *Whilst some simplification should be expected in a 'virtual' study like this, it is important not to over-simplify to the point where the study itself is so far removed from any actual application that the results are not comparable.* The problem here is that the authors consistently conflate their results with those from actual pollen-climate reconstructions (as in the title), and therefore are at risk of presenting a fallacious argument that the average reader who is not so familiar with the topic will likely interpret at face value.

We fully agree that the complexity of the vegetation representation in the model as well as the simulated climate evolution are a strong simplification of the reality. Therefore, results on the Holocene evolution of specific PFT's, the actual spatial pattern of PFT's, or the reconstructability of a certain climate variable in a certain region should not be directly translated to the real-world.

On the other hand, conclusions about reconstruction methods and the relation of spatial calibration and downcore reconstruction only require a consistent dataset of climate and vegetation parameters in space and time and do not depend on details of the climate evolution or vegetation response, as long as the dataset is realistic enough that we can apply the real world reconstruction workflow. The major factor shaping these results is that the modern spatial relationships between climate variables is different from the changes in the relationships over time, which is a robust feature related to the transient insolation forcing.

In the revision, we will check in detail again if all our statements are either independent from the model-world specifics , or are clearly marked that they just apply to the model world. Furthermore, we will emphasize the limitations of our study further by extending paragraph 4.1, and by highlighting their impact in the conclusion paragraphs. We do, however, find the title of our manuscript appropriate, as it clearly expresses that we work with pollen-related methods in an ideal model world.

Implemented changes in the revised manuscript:
- **Throughout the manuscript, in particular in paragraph 4.1 on limitations and the Conclusions we have evaluated and highlighted in a better way, which statements pertain to the model world experiment here, and which can be generalized to real-world pollen-based reconstructions.**

**Point 5: Repeating the study with a more elaborate vegetation model (LPJGuess)**

> The subject of the paper is nevertheless interesting, and one that would otherwise be worthy of publication. I would therefore encourage the authors to collaborate with someone who has more experience in pollen-climate modeling, and to use a vegetation model such as LPJGUESS which can simulate a greater number of PFT's/Taxa so that the analysis can be more comparable with how pollen-climate transfer functions are actually applied.

As we have shown above, our analysis is comparable to actual pollen-climate transfer functions, as the

effective number of species present in the model data is within the range of the numbers observed for the European Pollen Database, which formed the basis e.g. for (Mauri et al., 2014; Mauri et al., 2015).

We agree that using a more complex vegetation model, such as LPJGUESS, would be worthwhile in a future study as this would not only increase the number of PFT's but also include a more realistic interaction between climate and vegetation. This would allow to also test other properies of the reconstruction e.g. effects on the time-scale dependent variability, or potential time-lags. We will thus include this proposal in the outlook of our study.

However, we expect that the main result of this manuscript, the limitations of spatial modern calibrations, would only be strengthened. This is because a more realistic climate-vegetation response will even differ more in time versus space, if for example the modern vegetation is not yet in equilibrium with the modern climate state.

**Implemented changes in the revised manuscript:**
- **We state that it would be worthwile to repeat the study with a vegetation model with more realistic vegetation and more PFTs (p. 18 l. 16-25)**

- **We emphasize the limitations of our approach (p. 15 l. 2- p. 16 l. 17)**

**Response to anonymous reviewer #2**

**Summary**

We would like to thank the reviewer for her/his insightful remarks, which will help us to improve a revised manuscript.

Reviewer's comments are given in grey. Emphasis in *italics* was added to highlight main points.

**Point 1: Combination of climate parameters governing vegetation composition in the past**

The paper by Rehfeld et al. deals with the pollen-based climate reconstructions. The authors use climate model data and modelled vegetation to explore the reliability of reconstructions of different climate parameters in pollen-based reconstructions. The advantage in such an experiment "in an ideal model world" is that the past climate and vegetation are known at all times (6 ka to present), allowing to assess the reliability of the reconstructions. *The authors show that reconstructing multiple climate parameters can be misleading, as it is possible that in reality there is only one climate parameter which drives the spatial and temporal vegetation change, and the reconstructions of other climate parameters show temporal variability which is caused by the fact that these less important parameters are spatially correlated with the important parameter in the modern spatial data used for constructing the transfer function.* This is certainly nothing new, most of the palaeoecologists using pollen data have been aware of this problem, but it is useful to have a special study where this problem in explicitly explored using novel

approaches.

I find it easy to agree with the authors that "the temporal changes of a dominant climate variable are imprinted on a less important variable, leading to reconstructions biased towards the dominant variable's trend" and that the high r2 in the cross-validation is of limited use to identify which variables can be reconstructed, as r2 can be high not only for the variable which is really important for vegetation or pollen, but also to non-important variable which are spatially correlated with the important variable. The authors suggest assessing the amount of fossil vegetation variance explained the reconstruction output and expert knowledge as possible means to select the climate variables. *The latter one has been used in pollen-based reconstructions, but unfortunately the expert knowledge almost invariably is limited to present ecological setting.* It is possible, or even likely, that *if we go back in time enough, the combination of climate parameters governing the vegetation composition have been fundamentally different from the present.*

Non-analogue combinations of climate and other physiologically relevant variables certainly occurred in the past and will result in less accurate reconstructions. For example, the effect of low-CO2 concentrations on LGM pollen-based reconstructions is difficult to quantify and is hence largely ignored. Climatic conditions in the Holocene, the period analysed in this study are unlikely to have been sufficiently non-analogous to cause serious problems, and non-equilibrium vegetation may be more of a general problem.

**Point 2: Number of PFTs**

There is one striking problem with the paper. Given that the authors use model data only, they are restricted to use plant functional types (pft), not pollen types or plant species. In the real world, the WA-based climate reconstructions often comprise over 100 pollen types, not pfts. Modern analogue-based reconstructions use pfts, but even in them the number of pfts is generally 20-30. In a striking contrast, the number of pfts in the current study is eight - in other words extremely low. I am surprised that the palaoeclimate reconstructions with such a low number of variables make any sense in the first place, given that they are based on a few, extremely broad pft classes.

We thank the reviewer for raising this point. We agree that the number of  Plant Functional Types (PFTs) in our model study is lower than what is generally used in a real-world large-scale reconstruction exercise (eg. in Davis et al., 2003, Mauri et al., 2014, Mauri et al., 2015). We use 9 PFTs (one of which is representing bare soil, or desert fraction), whereas e.g. Mauri et al. (2014) use 22 (two of which are virtual). However, as pointed out in the response to Reviewer 1, what is ultimately relevant for the calibration and reconstruction efforts is the information contributed by the PFTs or taxa; in other words how many of them actually contribute to the pollen (or PFT) diagram in a relevant way. This number is much lower than the number of PFTs, or the number of taxa in a pollen diagram, but as we outline below, this number is comparable between our modeled PFTs and real-world pollen diagrams.

[Figure]

*Figure R3: Species response curves for the calibration radius around example site 120E,72N. Only non-zero taxa are shown.*

The median palynological richness in the 4990 sites in the European Pollen Database is 31. However, the median effective number of species (given by Hill's N2 (Hill, 1973)), discounting very rare taxa which would not be relevant for reconstructions, is 4.9. Of the 458 grid points we use, the median N2 for the fossil data (shown in Fig. 1d in the manuscript) is 2.9, and for the modern calibration data it is 2.7. Therefore, although the number of PFTs is much lower in the model, the diversity and effective number of species is not much lower than that many actual pollen-climate reconstructions. Moreover, we are setting bounds on N2 and turnover to avoid pathological problems due to too few taxa. This is an important aspect we will discuss in more detail than before in a revised manuscript.

**Implemented changes in the revised manuscript:**
- **inserted new discussion and definition of Hill's N2 on p. 5 l. 19.**

[Figure]

*Figure R4: Ratio of the standard deviation of the MTWA climate variable at modern analog sites over the standard deviation within the training set.*

- **main text paragraph p.5 l. 30 – p. 6. l. 10 on N2 vs. richness**

- **new Supplementary Figure SFig. 2**

- **We emphasize the limitations of our approach due to the low number of taxa (p. 15 l. 2- p. 16 l. 17)**

**Point 3: Multiple Analogues**

I suspect that the reconstructions using modern analogue must have included some serious problems which are not reported in the paper. The *problem of multiple analogues* (where the many modern analogues for the fossil sample are present, often in very different climatic settings) would be unavoidable with eight pdfs only.

We thank the reviewer for raising this important point. The multiple analog problem could arise if the species response curve (e.g. with respect to the climate variable, e.g. MTWA) within a modern calibration radius was multimodal. However, analyzing the species response curves at several sites suggests that this is not the case. As an example, Fig. R3 shows the species response curves for all taxa effectively present in the Siberian site for which we also show the complete reconstruction workflow in the manuscript (Fig. 2). The species response curves are not multimodal.

Furthermore, the overall high transfer function r2 (Fig. 7 in the manuscript) shows, that analogs are not picked at random from the training set, and underlines that multiple analogs are not a problem . To exemplify this we calculate the ratio of the standard deviations of the temperatures at the analog sites, and the standard deviation of the temperatures across the whole training sets (Fig. R4) . The ratios are generally smaller than 0.5, thus illustrating that the analog sites are not randomly drawn from the training set.

In the revised version of the manuscript we will explicitly demonstrate that arbitrary analogs are not a problem here, by including the above discussion.

**Point 4: Error estimates and uncertainties**

The *error estimates of the calibration sets and the fossil reconstructions are not presented or discussed in the paper, but they most likely are extremely high*. I therefore wonder if the *difference in the reconstructions (in Fig. 8 and 9, for example) are inside or outside the error estimates*?

For the sake of brevity we have not given explicit plots of all quality metrics within the manuscript. We do, however, give standard error estimates (RMSEP, cross-validation r2) for the example grid point reconstruction workflow (Fig. 2 in the manuscript), and for all grid points >50N (Table 1). The calibration r2 for all climate variables is given in Fig. 7 in the manuscript, and Fig. 7 in the supplement compares the different error estimates for downcore RMSE and RMSEP for warmest month temperature. While we have given the summary numbers for the RMSEP and the r2 in Table 1, we agree that it would be helpful to show these statistics explicitly for each gridpoint, projected onto a map. We show the results for the temperature variables below in Fig. R5. As suspected by the reviewer, the RMSEP, particularly for MTCO, is not small, however, it is not much larger than that in real-world large-scale reconstructions (c.p. Frechette et al., 2008; Mauri et al., 2014). The regions with high RMSEP are largely consistent with the regions where the calibrations' r2 is low (Fig. 7 in the manuscript). We will improve Figs. 4 and 5 in the manuscript by masking grid points with an r2 below 0.5, as the results show the covariance across time and space of winter and summer temperature reconstructions, and could be affected by this. Furthermore we plan to incorporate Fig. R5 in the supplement of the revised manuscript, and we will also better explain and more prominently discuss Table 1 in the revised manuscript.

[Figure]

*Figure R5: RMSEP for 10-fold-leave-group-out cross validation of MTCO (a), MTWA (b) and MAT (c). Black gridcells indicate a RMSEP larger than 5C.*

The question whether or not the temperature changes against time shown in Fig. 8 and 9 in the

manuscript are significant or not using the calibration RMSEP is not straightforward. A standard assumption in paleoclimate reconstructions is that errors in time and space are independent (as assumed e.g. in Marcott et al. 2013, Fedorov et al., 2013, Shakun et al., 2012). This assumption would result in a standard error of 0.008C (1se) for the difference of -0.72C between the calibration at 0k vs. 6k in Fig. 8 (taken across all 27 gridpoints >70N and 197 time points). For Fig. 9 the difference at the 0k is 0.32C, at 6k it is .44C and the standard error at these time points is smaller than 0.13C. Both differences in the reconstructions would therefore be highly significant. On the other hand, there are good reasons to expect spatial and temporal correlations in the errors. In the (unrealistic) extreme case of a complete dependency or errors, the differences would be not significant. In reality the true uncertainty likely lies between the two extremes assumed here but a mechanistic understanding of processes causing the proxy uncertainty is required to provide better error estimates. In the revised manuscript, will discuss these aspects of uncertainty in Section 4 and include the uncertainty estimates for both cases.

**Implemented changes in the revised manuscript:**

- **new Supplementary Figure SFig. 6, discussed in the manuscript text on p. 10 l. 7-9.**

- **enhanced Table 1 & improved linkage to the text**

- **masking of grid points with low transfer function performance (Figs. 4 & 5 in the manuscript)**

- **new discussion of uncertainties (for Fig. 8 & 9) on p. 14 l. 6-11.**

[revised manuscript text omitted]

---

## Author Response (AR2)

**Author response to reviewer and editor comments**

**Manuscript: "Assessing performance and seasonal bias of pollen-based climate reconstructions in a perfect model world" by K. Rehfeld, M. Trachsel, R.J. Telford and T. Laepple (doi:10.5194/cp-2016-13)**

**11/26/16**

**Response to editor comments**

We thank the editor for his comments. As suggested, we have improved the wording of the manuscript, following the third reviewer's points. The new version thus features an expanded introduction and conclusion and corrections throughout the text. This has substantially improved the readability of the manuscript.

**Summary of changes**

- extended introduction
- extended discussion
- augmentation of Fig. 2
- correction of spelling

**Response to referee #3, John Williams**

**Summary**

We would like to thank you for your detailed and constructive comments. Following your suggestions we have expanded the introduction, discussion and conclusions, modified two figures and corrected grammar.

**Point by point response**

Reviewer's comments are given in grey. Emphasis in *italics* was added to highlight main points.

**Point 1: Expansion of the introduction**

> **Reviewer's comments:**
> This is an important and provocative paper, presenting an interesting and innovative analysis that offers a fresh take on a long-standing problem**.** Specifically, whether the *paleoclimatic reconstructions that employ transfer functions*, modern calibration datasets, multivariate fossil assemblages, *and space-for-time substitution are heavily biased by assumptions of stability in the correlation structure among climatic variables in space and time.* Prior papers by these authors and others (e.g. Juggins 2013) have argued that shifting correlation structures among variables can lead to major and often unrecognized biases.
>
> This paper makes a new contribution by working entirely with a set of modeled climates and vegetation (PFTs), for which the actual climate evolution for the Holocene is known. Hence, the authors can apply standard transfer function methods to the modeled vegetation PFTs and see how well the reconstructed climates compare to the simulated climates. The results clearly show that in this modeled context, the PFT-based climate reconstructions are really only able to reconstruct one variable (MTWA) well, in the focal region of Siberia.
>
> This paper is timely given the recent reconstructions of Northern Hemisphere mean annual temperature by Marcott et al (2013) and the challenge by Liu et al. (2014) that the pollen-based paleotemperature reconstructions primarily represent summer instead of winter temperatures.
>
> I've read both the paper and the commentary and found it all fascinating. My overall assessment is strongly for publication –this is the kind of paper that will spark debate but I think in a net positive direction; it should move the conversation forward. However, the paper can be expected to meet resistance from the proxy-based paleoclimatic community, who can easily dismiss it along the grounds outlined by the original two reviewers: the model is very simple, it contains only 9 PFTs, etc. I personally am not entirely convinced by the authors' response. I personally find the findings more illustrative than definitive, given the simplicity of the vegetation model and its PFTs, and think that the paper should present its findings as such.
>
> Hence, to really have the impact it deserves, t*his paper needs to further strengthen its introduction discussion. A careful weighing of the paper's own limitations and caveats* will go a long way to strengthening the paper's overall argument and moving the conversation forward.

We thank the reviewer for this balanced evaluation and encouraging comments. We have extended the introduction, discussion and conclusions to make the limitations, caveats and possibilities of this manuscript, and future studies more clear.

**Point 1: Expansion of the introduction**

> 1. **Introduction**: This should be expanded, to better set relevance and context. Specifically:
>
> a. It should *mention the space-for-time assumption* and clearly establish the underlying assumption that variables correlated in space also correlated in time. Explicitly state that we have a good a priori reason to expect this assumption to be violated for temperature, given a) the strong correlation of summer and winter temperature in space today and b) the known anticorrelation of summer and winter insolation in the NH over the past 10,000 years due to precessional forcing. (this is mentioned very late in ms., on P15L3-5)
>
> b. Could more fully *describe Juggins 2013* – this paper is cited in passing, but its key point about confounding variables isn't really explained.
>
> c. Cite the *Marcott et al. and Liu et al.* papers in the intro, as a way of signaling the importance of this topic in current data-model comparisons and conversations about Holocene warmth vs. 21st-century temperature changes. This topic is introduced in Discussion, but should be introduced earlier.

We have extended the introduction following the reviewer's suggestions. In the revised manuscript, we use a more nuanced treatment of uniformitarianism and the space-for-time substitution is discussed in more depth. The model-data discrepancy for Holocene temperatures, and the different trends expected for summer and winter temperature due to the insolation changes through the Holocene are now mentioned earlier in the manuscript.

**Point 2: Discussion of Uniformitarianism**

> 2. *Uniformitarianism* is appropriately a central theme of this paper. However, there are several distinctly different kinds of uniformitarianism. Lyell founded the discipline of geology by assuming that processes observable today also operated in the past. Assuming uniformitarianism of processes is fine. However, Lyell also assumed that rates were roughly uniformitarianism through time, which is false. Gould called these two forms of uniformitarianism 'methodological' (uniformitarianism applied to processes, true) and 'substantive' (uniformitarianism applied to rates, false) (http://philpapers.org/rec/GOUIUN). This paper is dealing with a third kind of uniformatiariansm, assuming that covariance structures are uniform through time. This, like the 'substantive' uniformitarianism, is clearly false.
>
> a. So, I strongly suggest a more nuanced treatment of uniformitarianism that distinguishes these concepts.
>
> b. Also, remove quotes around this phrase.
>
> c. Adding and defining a new phrase ('**correlative uniformitarianism'?**) can give others an easy way of citing the arguments in this paper.

We appreciate the reviewer's remarks and have expanded the introduction section. We specifically now only refer to the methodological uniformitarianism and introduce the expression 'correlative uniformitarianism' to describe the (violated) assumption of covariance structures that are uniform through time and equal to the spatial covariance structures. We hope that the new introduction makes it easier to understand the main challenges.

**Point 3: Expansion of the discussion**

> 3. In *Discussion, strengthen your case by pointing to other papers that have also explored the stability of correlation structures and the effect of confounding variables.*
>
> a. Salonen et al. 2013 Holocene for demonstrating effect of alternate calib datasets and continentality on . TJul reconstructions.
>
> b. Blois et al. 2013: explicitly tested the space-for-time assumption by running generalized dissimilarity models on spatial vs. temporal datasets of species turnover. They argued that GDMs fitted across space could predict emergent patterns of diversity across space, but also found that the modeled relationships between diversity and turnover varied quite a bit among climate variables.
>
> c. More clearly set up the issue of assuming stable correlations and the issue of secondary variables. Insolation. See P15L3-5.
>
> d. Should acknowledge that Siberia may be an end-member/worst-case region for paleoclimatic transfer functions, in which summer light and warmth is really critical. Hence, this may be a worst-case system for multivariate transfer functions where one variable really dominates (MTWA) and the others are very secondary. In other regions, multiple climatic controls on vegetation may be important and disentanglable by transfer functions.

We have followed the reviewer's suggestion and have expanded the discussion section. We added a new dicussion section 'Correlative uniformitarianism' that discuss our findings in the light of the suggested references. Further, we extended the discussion of insolation as a latent variable both in the Introduction and in the Discussion sections. We agree that the Holocene evolution of vegetation in Eastern Russia is primarily dominated by summer length and temperature changes, and in other regions other variables dominate (c.f. Fig. 6).

We have largely refrained from discussing other pairs of variables (e.g. temperature/precipitation) for simplicity. We have rephrased sentences in the Abstract and Discussion to make it more clear that we do not expect summer temperatures to dominate everywhere and added a paragraph in the section Limitations that our Arctic Russia case is an end-member concerning the domination of MTWA.

**Point 4: Discussion of the role of N2**

> **4.** *I am not convinced by the N2 analyses presented by the authors. I have two major concerns:*
>
> a. The paper never establishes what is a meaningful difference in N2. It simply states that the N2 in pollen data and the N2 in the modeled PFTs are about the same. However, I suspect that for paleoclimatic transfer functions, even a difference of 0.5 in N2 could be important, given that this represents in some sense the degrees of freedom available to the transfer functions. So if e.g. N2 is ~1.5 for PFTs and ~2.5 for the pollen data, that would imply an extra degree of freedom or so in the pollen data and more multivariate power.
>
> b. For any given time or place in the pollen data, N2 might be about 2, but over the entire global region, I'd suspect that N2 is >>2. But in the model, Global N2 can never be higher than 8, and in the extratropics, can only be 5.

We thank the reviewer for his comments on the effective number of species/plant functional types.

To our knowledge, there is no literature on meaningful differences in N2. It is therefore difficult to give a conclusive statement on meaningful differences in N2. Our goal here was to highlight that the difference in N2 of PFTs between 'model world' and 'real world' is smaller than the difference in apparent number of pollen taxa or PFTs. The question of meaningful differences in N2 could be addressed in further studies, and we have explicitly added them to the paragraphs on future work.

We would expect a reconstruction with higher N2 to be more reliable and also to improve the possibility to reconstruct more than one climate variable, however, this does not change the problems with spatial vs temporal correlation ('correlative uniformitarianism' as suggested by the reviewer or 'sick science' (Juggins 2013)). Regarding the potential to simultaneously reconstruct multiple climate variables from pollen assemblages, transfer functions (especially BMA) are probably too simple to make the inversion of pollen = f(MTWA, MAP asf.)

Regarding N2, we also notice that reconstructions use PFTs instead of pollen types. Thereby a reduction in N2 is implicitly accepted, while at the same time the relation between PFTs and climate is potentially getting more stable.

For this study, N2 at global scale is not that relevant for a number of reasons:  In the 'real' as well as in the 'model' world, all calibration data sets used are on a regional to continental level.

All reconstructions are based on local pollen and are therefore limited by N2 in the fossil (local) data set (having a higher N2 in the calibration data set is not changing N2 in the local data set). The N2 relevant for a reconstruction is ultimately the N2 of the fossil pollen/PFTs. As this study is about climate reconstructions based on pollen/PFTs, the N2 at global scale is not directly relevant in this study.  Vegetation changes occurring during the 6000 years are not large enough to make, for instance, tropical plants relevant as analogs for reconstructions in temperate climates.

In the revised manuscript, we discuss the difficulty of a meaningful difference in N2 in the Limitations

section and in propose in "Future work" section that the impact of species richness on the reconstruction skill should be explored.

**Point 5: Expansion on caveats and future work**

> 5. In *Abstract and Conclusions:* Add caveats. Note that number of PFTs of study are fewer than in modern pollen datasets. But nevertheless, the issues raised here about confounding variables are consistent with those from empirical studies of calibration datasets.
>
> a. In Conclusions, add a 'more work is needed' sentence with a pointer to a LPJ-GUESS study. Well posed 'future work' statements can be very effective at moving the field forward and spurring future work, either by the authors or by other teams.

Following the suggestions we have re-worded parts of the Abstract, the Discussion, Outlook and the Conclusions to make it more clear that we consider this work as an idealized study which could be expanded in several ways.

**Point 6: Corrections**

> **MISCELLANEOUS COMMENTS**
>
> Summer Temperature. One takeaway message of this paper by a naïve reader could be that summer temperature (MTWA) is the most critical variable, and so pollen-based paleoclimatic reconstructions should restrict themselves to MTWA. The abstract itself implies that summer temperature is the critical variable. However, the rest of the paper shows more complexity and caveats to this inference:
>
> • These are modeled results, and the model may be more sensitive to summer temperature than real-world vegetation.
>
> • Figure 4 shows interesting deviations from this, particularly in the tropics and subtropics, where MTCO seems to be better reconstructed than MTWA.
>
> • Fig 6 shows that the variables explaining variance in vegetation vary regionally, e.g. MAP in the tropics.
>
> • Siberian focus really emphasizes MTWA, as noted above.
>
> *I suggest adding nuance to the abstract and adding a section of the Discussion specifically focused on the question of whether MTWA is always the best variable for pollen-based vegetation reconstructions. That would directly speak to the discussion by Liu et al. and*

*Marcott et al.*

We have carefully checked the manuscript to ensure that it is most clear that our statements, insofar as they concern the reconstructability of individual climate variables, are not misleading. Given that these are results based on a model, as noted by the reviewer above, vegetation may be more sensitive to summer temperature changes in the Arctic than in the real world. We discuss the regional patterns of driving variables in Fig. 4, and have added a paragraph in the Discussion section on future work that could be done to identify reconstructible variables.

Throughout paper, be very careful to not mix up PFTs and species – they are very different. For example, when referring to Hill's N2 for the model simulations, use 'effective number of PFTs' . 'Taxa' also would be a good option.

We have changed the wording following the reviewer's suggestion.

P10L5: A RMSEP of 3C would be mostly unacceptable in real-world Holocene paleotemperature reconstructions, given that the Holocene signal of temperature change is on the order of 1-2 degrees in many places. Fig. 8 shows a similar trend, on the order of 2C. Adjust wording and note that this may be the case where the PFT-based paleoclimatic transfer functions are doing much worse than real-world pollen-based transfer functions.

We have removed the "acceptable" statement. What we want to state at this point of the manuscript is that the RMSEP is lower for MTWA than MTCO in the model world. It is, however, not clear (to us) if a higher RMSEP is a general feature of PFT-based palaeoclimate reconstructions (e.g. Mauri et al.,2011).

P14L28-30: 'real world' is vague. Clarify that what you mean is that these results may not be applicable to pollen-based paleoclimatic transfer functions, because of their higher richness.

We have expanded the "Limitations" section and clarified that we do not expect the simulation results to reflect actual vegetation or climate changes through the Holocene. We pose the question how high a Hill's number has to be to ensure reconstructability and come back to this point in the "future work" paragraph.

Figure 1: Clarify that these maps are from the model simulations. Fix axis title that says "# species" – it should say "# PFTs" or "# effective PFTs"

Corrected.

Figure 2:

This figure packs in a lot of information. It needs more information in axis titles and legend.

Fig 2B: Not clear that MTWA explains the most variance in modern vegetation, as claimed in Fig. legend.  This is inferred from Fig. 2D. We have made the statement more clear.

Fig 2C: Clarify axes – is this 'temperature' MTWA, MTCO, or MAT?  The axis label has been corrected to read  "MTWA temperature".

Figure 2D: Vertical axis? What is this a % of? The axis denotes the proportion of variance explained by the different variables. We now spell this out on the vertical axis.

Figure 2E: Define the PFT acronyms in legend.  We have added a reference to the appendix where we list the PFT acronyms. For space reasons we prefer not to give the acronyms in the figure caption or legend. We noted that we had omitted the surrogate PFT "bare soil" in the appendix and have added this to the list as well.

Figure 2G: What is 2G? It's not mentioned in legend. Vertical axis title? What are the dashed vertical lines?  We have added the missing "(2G)"  before "The MTWA reconstruction explains most fossil vegetation variance ..." in the legend and added an explanation for the vertical lines, which are derived from the randomTF algorithm (Telford and Birks, 2011).

Figure 7, exploring R2 – seemed less critical – could delete this figure. We prefer to keep Fig. 7 as it illustrates how several transfer functions can have a near-equal transfer function R2, to the point where it becomes irrelevant as a predictive diagnostic for transfer function performance.

[revised manuscript text omitted]